# A Serial Cross-Sectional Analysis of the Prevalence, Risk Factors and Geographic Variations of Reduced Visual Acuity in Primary and Secondary Students from 2000 to 2017 in Hong Kong

**DOI:** 10.3390/ijerph17031023

**Published:** 2020-02-06

**Authors:** Perseus W.F. Wong, Jimmy S.M. Lai, Jonathan C.H. Chan

**Affiliations:** Department of Ophthalmology, L.K.S. Faculty of Medicine, The University of Hong Kong, Hong Kong, China

**Keywords:** adolescent health, community health, geographic disparities, visual acuity, time-series surveillance, retrospective analysis

## Abstract

*Background:* We would like to investigate the prevalence trend, potential risk factors and geographic features of reduced visual acuity (VA) in primary and secondary schoolchildren of Hong Kong. *Methods:* This was a serial cross-sectional study using historical data of schoolchildren aged 6 to 15 years from the annual health checks conducted at Student Health Service Centers across Hong Kong, for the school years of 2000/2001 to 2016/2017. *Results:* The prevalence of reduced VA increased from 49.23% (95% CI, 48.99−49.47) in 2000/2001 to 54.34% (95% CI, 54.10−54.58) in 2011/2012 but decreased to 51.42% (95% CI, 51.17−51.66) in 2016/2017. Girls were less susceptible than boys at age 6−7 (and in grade primary 1−2), but more susceptible at older ages. The prevalence in junior grades increased while the risk effect of grade reduced over the past 17 years. Geographic variation on the risk for reduced VA existed and spatial autocorrelation was positive. The difference in prevalence of reduced VA between Hong Kong and mainland China has decreased in recent years. Cross-border students living in mainland China were associated with a lower risk for reduced VA. *Conclusions:* Further study was proposed to investigate the environmental association between students living in and outside Hong Kong with the prevalence of reduced VA. Multi-level research should also be conducted to investigate the influence of compositional and contextual factors on the prevalence of reduced VA.

## 1. Introduction

Refractive error was the leading cause of reduced visual acuity (VA) among children and adolescents worldwide. Studies of this century in Vietnam, India and Malaysia have found that refractive error accounted for 80% to 96% of reduced VA [1,2,3]. In mainland China, refractive errors accounted for 85% to 98% of reduced VA in Beijing, Chongqing, Xinjiang, Yunnan, Guangdong and Shanghai [4,5,6,7,8,9,10]. One common cause of refractive errors among schoolchildren was myopia, accounting for 25% to 75% of refractive errors in Saudi Arabia, Nepal, India, Norway and Mexico [11,12,13,14,15]. Myopia accounted for 53% to 67% of refractive errors in Shanghai, Chongqing and Beijing of China [7,9,16].

To our knowledge, three surveys were done for estimating the prevalence of refractive errors or myopia among schoolchildren in Hong Kong (HK). A 2005−2010 survey by Lam et al showed that the prevalence of refractive errors was 34.2% and 66.6% at age 6 and 10, respectively [17]. Another survey found that the prevalence of myopia in local and international schoolchildren in 2001 were 85−88% and 60−66%, respectively, with the highest in the Chinese group (82.8%) and the lowest in the white group (40.5%) [18]. The third survey noted that the prevalence (± standard deviation, SD) of myopia at age 5−16 was 36.71% (±2.87%) from 1998 to 2000 [19]. However, they had three limitations. 

First, their sample size was small because large-scale adoption of cycloplegic refraction was not feasible. In mainland China, two retrospective serial cross-sectional studies were conducted to estimate the prevalence of reduced VA in large populations, assuming that reduced VA would be a proxy measure for myopia [20,21]. Their rationale was based on the findings of three previous studies of testing the sensitivity and specificity of using unaided VA to screen for refractive errors among students in Singapore and Australia [22,23,24]. 

Second, they did not measure geographic variations on the risk for reduced VA and spatial autocorrelation in HK. Previous studies proved the associations of areas of residence (urban/rural region), housing types (flat room/building), living environment (frequency of seeing green) and economic status (socioeconomically advantaged/disadvantaged) with myopia, one major cause of refractive errors [25,26,27,28]. If these variables were similarly or dissimilarly clustered with the spatial location, there would be positive or negative spatial autocorrelation of reduced VA.

Third, they estimated the prevalence at a single time point but did not describe the dynamic changes over a series of time points, e.g., cross-boundary students with distinct sociodemographic features may have led to changes in prevalence over time. In mainland China, two retrospective serial cross-sectional studies were conducted to estimate the prevalence of reduced VA in Guangzhou, China from 1988 to 2007 and the whole of China from 1985 to 2010. In contest, no serial cross-sectional studies, whether prospective or retrospective, have been conducted in HK. 

For understanding the epidemiologic trend of reduced VA, we would like to review the historical data of Health Programs at Student Health Service Centers (SHSCs) of the Department of Health, Hong Kong Special Administrative Region (DH, HKSAR) [29]. The objective of our study was to investigate the prevalence trend, potential risk factors and geographic features of reduced VA in primary and secondary students from the school year 2000/2001 to 2016/2017. 

## 2. Materials and Methods

### 2.1. Study Area

The Student Health Service (SHS) of the DH, HKSAR has been offering annual health checks for primary and secondary students (including local and international schools) of HK since the 1995/1996 school year for primary students and the 1996/1997 school year for secondary students. This includes regular VA assessment, with the aim to detect any VA problems of students and whether these problems need referrals or have been appropriately corrected. Clinical data pertaining to visual assessment of these students were obtained from 12 SHSCs throughout HK.

The Hong Kong Special Administrative Region is divided into 18 administrative districts, which are: Central and Western, Wan Chai, Eastern, Southern, Yau Tsim Mong, Sham Shui Po, Kowloon City, Wong Tai Sin, Kwun Tong, Kwai Tsing, Tsuen Wan, Tuen Mun, Yuen Long, North, Tai Po, Sha Tin, Sai Kung and Islands [30]. 

### 2.2. Study Design

This is a retrospective, serial, cross-sectional study covering primary and secondary students aged 6 to 15 years attending the annual health check offered by the SHS from the school year of 2000/2001 to 2016/2017, with the exception of 2009/2010, since in that school year, SHS had to take part in the Human Swine Influenza Vaccination Program, and therefore, annual appointments were only provided to Primary 1 (P1) to Secondary 1 (S1) students.

### 2.3. Source of Data

Available records between 2000/2001 and 2016/2017 from the SHS database were extracted for relevant data; including age, sex, school type, grade of student, school district, location of SHSCs, type of housing, home district and VA test result. The relative age and sex composition of the study population was retrieved from the Student Enrollment Statistics of Education Bureau [31].

### 2.4. Definitions

VA testing was performed for distant vision using a Logarithm of Minimum Angle of Resolution (logMAR) VA display chart at 3 or 4 meters. Students with an unaided logMAR VA of 0.3 or better were recorded as having satisfactory vision. The test was done with visual aid if the students was wearing glass or contact lens. Reduced VA was defined as having an unaided logMAR VA of >0.3, or wearing glasses or contact lenses (excluding Ortho-K lenses). 

### 2.5. Data Analysis

We used multiple imputation to create 20 data sets with imputed values for handling the missing data. The means and odds ratios (ORs) were calculated by using SAS PROC SURVEYMEANS and PROC SURVEYLOGISTIC, respectively. We combined the outcomes of 20 datasets and provided the final estimates with 95% confidence interval (CI) in SAS PROC MIANALYZE. 

The prevalence of reduced VA was reported in crude rate, age-sex adjusted rate after adjusting the sampling weight of age and sex composition to poststratification weight for each school year, and standardized rate for selecting the study population for 2000/2001 as a standard population. We divided the sample data into 10 strata (5 age groups (6−7, 8−9, 10−11, 12−13 and 14–15) × 2 sex groups (boys and girls)), and combined them with the population data from Education Bureau to do poststratification analysis and direct standardization. Mann-Kendall trend test is a non-parametric test used to analyze the data for consistently increasing or decreasing trends in the series. We tested the null hypothesis that there was no monotonic trend for the prevalence of reduced VA over the study period. Subgroup analyses stratified by age (6−7, 8−9, 10−11, 12−13 and 14−15) and grade (P1-P2, P3-P4, P5-P6, S1-S2 and S3-S4) were also done to identify either consistency of or large differences in the prevalence and ORs of reduced VA among different categories of students.

We performed univariate logistic analysis with and without adjusting for age and sex and fitted all independent variables into multivariate logistic analysis to measure the strength of their association with reduced VA. The results were presented in four time intervals: 2000/2001−2003/2004, 2004/2005−2007/2008, 2008/2009−2012/2013 and 2013/2014−2016/2017. Chi-square independence test was used to test whether the presence of reduced VA and the 18 districts (home or school) were independent when Kruskal–Wallis test were used to test whether the prevalence of reduced VA were identical among the 18 districts (home or school). Mantel test was computed to test whether there was a relationship between the community distance from Central and Western District (where the city center is located) and the prevalence (or AORs) of reduced VA. Moran’s Index was calculated to test whether there was a spatial clustering of the prevalence (or AORs) of reduced VA associated with the community distance from Central and Western District and also measure the overall spatial autocorrelation. 

All data extraction, processing and analyses were performed using the R package version 3.5.1 (R Development Core Team, 2018) and SAS version 9.4 (SAS Institute Inc., Cary, NC, USA). The results were reported in estimate with 95% CI and with *p*-value < 0.05 being considered statistically significant. 

### 2.6. Ethics Approval 

Ethics approval was granted by the Institutional Review Board (IRB) and Ethics Committees (EC) of the Hong Kong University/Hospital Authority Hong Kong West Cluster (UW 18-071) on 11 January, 2018. Availability of data and computing code: due to the ownership, the original dataset would not be available without authorization from the Department of Health in Hong Kong but the computing code are available from the corresponding author on reasonable request. 

## 3. Results

### 3.1. Demographics

The overall frequency of schoolchildren meeting the inclusion criteria were 6.75 million from 2000/2001 to 2016/2017. On average, the frequency was 0.422 million per school year. The participation rate ranged from 55.73% in 2000/2001 to 69.76% in 2013/2014. The range of mean (±SD) age was from 9.72 (±2.49) in 2000/2001 to 10.64 (±2.73) in 2010/2011. The proportion of boys increased from 49.93% in 2000/2001 to 51.37% in 2016/2017. The mean (±SD) presenting VA of right and left eyes were 0.229 (±0.142) and 0.235 (±0.136), respectively, in 2000/2001 and changed to 0.173 (±0.18) and 0.171 (±0.175), respectively, in 2016/2017. The proportions of wearing glasses or contact lenses ranged from 32.43% in 2000/2001 to 42.9% in 2010/2011. The percentage frequency of missing data of three variables (i.e., home district, presenting VA and usage of visual aids) was almost less than 1%. (Appendix A)

### 3.2. Prevalence

The standardized prevalence of reduced VA rose from 49.23% (95% CI, 48.99–49.47) in 2000/2001 to 54.34% (95% CI, 54.10–54.58) in 2011/2012 but dropped to 51.42% (95% CI, 51.17–51.66) in 2016/2017. Its correlations with the proportion of primary to secondary students attending SHS and enrolled students were −0.875 and −0.885, respectively. Mann-Kendall trend tests showed that the prevalence trend was rising from 2000/2001 to 2011/2012 (tau, 0.927; *p*-value, <0.001) but was null from 2012/2013 to 2016/2017. The overall prevalence trend remained increasing (tau, 0.383; *p*-value, 0.043) (Table 1).

Stratified by age, the standardized prevalence rose from 26.08% (95% CI, 25.76−26.40) in 2000/2001 to 31.13% (95% CI, 30.70−31.55) in 2011/2012 but dropped to 24.47% (95% CI, 24.13−24.81) in 2016/2017 for age 6−7. Moreover, it rose from 66.22% (95% CI, 65.37−67.07) in 2000/2001 to 69.46% (95% CI, 68.81−70.10) in 2011/2012 but dropped to 68.89% (95% CI, 68.14−69.63) in 2016/2017 for age 14−15. Mann-Kendall trend tests showed that the prevalence trend for all age groups was rising from 2000/2001 to 2011/2012 (*p*-value, <0.001) and was null from 2012/2013 to 2016/2017, except age 12−13 (tau, 1; *p*-value, 0.027). The overall prevalence trend was increasing for age 10−13 only. (Appendix A)

Stratified by grade, the standardized prevalence rose from 27.40% (95% CI, 27.11−27.70) in 2000/2001 to 33.76% (95% CI, 33.35−34.18) in 2011/2012, but dropped to 27.36% (95% CI, 27.01−27.70) in 2016/2017 for grade P1-P2. Moreover, it rose from 69.05% (95% CI, 68.06−70.05) in 2000/2001 to 70.48% (95% CI, 69.81−71.16) in 2011/2012, but dropped to 69.44% (95% CI, 68.67−70.20) in 2016/2017 for grade S3-S4. Mann-Kendall trend tests showed that the prevalence trend for all age groups was rising from 2000/2001 to 2011/2012 (*p*-value, <0.001) and dropping from 2012/2013 to 2016/2017 for grade P3-P4 (tau, −1; *p*-value, 0.027). The overall prevalence trend was increasing for grade P3-P6 only. 

### 3.3. Age and Sex

Overall, girls have higher risk for reduced VA relative to boys with stable OR of around 1.1 throughout the study period in univariate and multivariate logistic analysis. 

If stratified by age, girls were less susceptible to reduced VA at 6−7 years but became more susceptible at 8−9 years from 2000/2001 to 2008/2009 and at 10−11 years from 2010/2011 to 2016/2017. In 2000/2001, the ORs of girls were 0.937 (95% CI, 0.911−0.964) for age 6−7, 1.033 (95% CI, 1.008−1.057) for age 8−9 and 1.175 (95% CI, 1.124−1.228) for age 14−15. Ten years later, in 2010/11, the ORs of girls were 0.928 (95% CI, 0.898−0.960) for age 6−7, 1.126 (95% CI, 1.097−1.156) for age 10−11 and 1.157 (95% CI, 1.123−1.193) for age 14−15 (Appendix A).

If stratified by grade, girls were also less susceptible to reduced VA at grade P1–P2 but again became more susceptible at grade P3–P4 from 2000/2001 to 2010/2011 and grade P5–P6 from 2011/2012 to 2016/2017. In 2000/2001, the ORs of girls were 0.957 (95% CI, 0.933−0.982) for grade P1–P2, 1.046 (95% CI, 1.023−1.070) for grade P3–P4 and 1.169 (95% CI, 1.111−1.231) for grade S3–S4. Just over 10 years later, in 2011/2012, the ORs of girls were 0.929 (95% CI, 0.902−0.957) for grade P1–P2, 1.124 (95% CI, 1.094−1.155) for grade 5−6 and 1.215 (95% CI, 1.173−1.259) for grade S3–S4.

The result showed that an interaction effect between age (or grade) and sex was statistically significant. Sex would modify the effect of age (or grade) on reduced VA. Students with older age (or more senior grade) were associated with a higher risk of reduced VA, regardless of their sex. Being a girl meant a fixed reduction of risk regardless of age (or grade), but with an additional incremental risk according to the interaction term between age (or grade) and sex. The net effect of being a girl (compared to being a boy) was initially negative, but subsequently became positive, with regard to the risk of having reduced VA. For example, in 2000/2001, relative to boys at age 6, the corresponding ORs of boys at age 7, 8 and 15 were 1.163, 1.613 and 5.525, respectively. Compared to boys of the same age, girls have a lower risk of reduced VA at age 7 (0.893 × 1.095 = 0.978), but higher risk from age 8 (0.893 × 1.151 = 1.028) to 15 (0.893 × 1.361 = 1.215). Similarly, relative to boys at grade P1, the corresponding ORs of boys at grade P2, P3 and S4 were 1.292, 1.805 and 6.827, respectively. Again, girls have a lower risk of reduced VA than boys at grade P2 (0.909 × 1.085 = 0.986), but higher risk from grade P3 (0.909 × 1.137 = 1.034) to S4 (0.909 × 1.273 = 1.157) (Table 2). 

Older age was significantly associated with an increasingly higher risk for having reduced VA. From 2000/2001−2003/2004 to 2013/2014−2016/2017, the crude odds ratios (CORs) of age increased from 1.251 (95% CI, 1.249−1.252) to 1.267 (95% CI, 1.265−1.268). (Appendix A) The ORs increased from 1.234 (95% CI, 1.232−1.237) to 1.330 (95% CI, 1.327−1.333) after adjusting for school type and sex and increased from 0.954 (95% CI, 0.950−0.958) to 1.095 (95% CI, 1.088−1.102) after adjusting for grade and sex. (Appendix A) In multivariate logistic analysis, the adjusted odds ratios (AORs) increased from 0.974 (95% CI, 0.969−0.979) to 1.097 (95% CI, 1.091−1.104) (Table 3).

### 3.4. School Type and Grade of Students

Relative to primary school, studying in secondary school had a higher risk for reduced VA with a decreasing trend of ORs from 2.898 (95% CI, 2.875−2.920) to 2.696 (95% CI, 2.676−2.716) before adjusting for age and sex, and a decreasing trend of ORs from 1.106 (95% CI, 1.094−1.119) to 0.707 (95% CI, 0.699−0.716) after adjustment from 2000/2001−2003/2004 to 2013/2014−2016/2017. The corresponding relative changes in OR were 162.03% and 281.33% in univariate analysis and were 117.48% and 191.47% in multivariate analysis. Age was a positive confounder and exaggerated the observed association between school type and reduced VA (Table 4). 

Senior students had a higher risk for reduced VA with an increasing trend before adjusting for age and sex, and a decreasing trend after adjustment from 2000/2001−2003/2004 to 2013/2014−2016/2017. Relative to grade P1, for example, the ORs of grade S4 increased from 7.930 (95% CI, 7.732−8.134) to 8.864 (95% CI, 8.673−9.059) before adjusting for age and sex, and decreased from 11.923 (95% CI, 11.370 to 12.504) to 4.025 (95% CI, 3.802−4.262) after adjustment. The corresponding relative changes in OR were −33.49% and 120.22% in univariate analysis and were −21.52% and 126.47% in multivariate analysis. Age was a negative confounder in 2000/2001−2003/2004, and a positive confounder from 2004/2005−2007/2008 to 2013/2014−2016/2017. 

### 3.5. Type of Housing

Relative to public rental housing, during 2000/2001−2003/2004, the CORs of living in subsidized Home Ownership Scheme (HOS) flats, private housing, villas and squatter were 1.133 (95% CI, 1.122−1.145), 1.270 (1.260−1.280), 0.717 (0.703−0.732) and 0.645 (0.626−0.664), respectively. Adjusting for age and sex, the corresponding ORs were 1.217 (95% CI, 1.204−1.230), 1.341 (1.330−1.352), 0.767 (0.751−0.782) and 0.561 (0.545−0.578). In multivariate analysis, the corresponding AORs were 1.153 (95% CI, 1.141−1.166), 1.203 (1.193−1.214), 0.792 (0.775−0.809) and 0.607 (0.589−0.626). During 2013/2014−2016/2017, the CORs of living in subsidized HOS flats, private housing, villas and squatter were 1.101 (95% CI, 1.087−1.115), 0.952 (0.944−0.960), 0.698 (0.685−0.710) and 0.772 (0.729−0.817), respectively. Adjusting for age and sex, the corresponding ORs were 1.078 (95% CI, 1.064−1.092), 0.990 (0.981−0.998), 0.720 (0.707−0.734) and 0.713 (0.672−0.757), respectively. In multivariate analysis, the corresponding AORs were 1.067 (95% CI, 1.053−1.082), 0.984 (0.975−0.993), 0.776 (0.761−0.791) and 0.745 (0.702−0.790).

### 3.6. School District and Home District

Throughout the study period, chi-square independence test (*p*-value, <0.01) showed that the presence of reduced VA and the 18 districts (school or home) were not independent and had a significant relationship. Kruskal–Wallis test (*p*-value, <0.01) showed that the prevalence of reduced VA was nonidentical among the 18 districts (school or home), and at least one of them was differentiated from the others in relation to the prevalence of reduced VA. 

Figure 1 and Figure 2 displayed the AORs for reduced VA relative to Central and Western District by the student’s school district and home district, respectively on the map of HK. By school district, there were 3, 6, 4 and 4 districts having AORs of 0.85−0.94, 0.75−0.84, 0.65−0.74 and <0.65, respectively, during 2000/2001−2003/2004; and 1, 1, 5, 9 and 1 districts having AORs of ≥1.15, 1.05−1.14, 0.95−1.04, 0.85−0.94 and 0.75−0.84, respectively, during 2013/2014−2016/2017. By home district, there were 1, 12 and 4 districts having AORs of 1.05−1.14, 0.95−1.04 and 0.85−0.94, respectively, during 2000/2001−2003/2004, and 4, 6, 6 and 1 districts having AORs of 0.95−1.04, 0.85−0.94, 0.75−0.84 and 0.65−0.74, respectively, during 2013/2014−2016/2017.

Table 5 showed that in case of age-sex adjusted prevalence, the correlation coefficients of Mantel test were 0.341 and 0.426 for home district in 2008/2009−2012/2013 and 2013/2014−2016/2017, respectively, and 0.406 for school district in 2013/2014−2016/2017. In case of AORs, the correlation coefficients of Mantel test were 0.385, 0.683, 0.636 and 0.285 for home district, while Moran’s Index was 0.368 in 2004/2005−2007/2008 and 0.336 in 2008/2009−2012/2013 for home district. We showed that the spatial autocorrelation was positive between the distance from home district to city center (at the Central and Western District) and the prevalence (or AORs) of reduced VA, that is, there would be clustering of home districts with similar estimates of prevalence (or AORs).

## 4. Discussion

### 4.1. Prevalence

Our results showed that the prevalence of reduced VA in 2010/2011 were 29.97% for age 6−7 and 58.06% for age 10−11. Inferring from previous reports in mainland China of refractive errors accounting for 85−97% cases of reduced VA [4,5,6,7,8,9,10], our estimated prevalence of refractive errors in Hong Kong is lower than the earlier report by Lam et al. of 34.2% and 66.6% for the respective comparable age groups [17]. In contrast to that study, the ethnicity of our subjects was not limited to Chinese, nor were the recruitment sites limited to just six specific primary schools located in Hong Kong. 

Prevalence of myopia from a previous survey in Taiwan was 21% and 61% at age 7 and 12, respectively [32]. In our study, the prevalence of reduced VA was 26.08% and 61.48%, at age 6−7 and 12−13, respectively, suggesting a lower prevalence of myopia compared to Taiwan, after making similar inference from the relationship between reduced VA and myopia as mentioned. 

Data from mainland China showed that the prevalence of reduced VA amongst schoolchildren aged 7 to 18 years were 38.5%, 49.5% and 56.8% in 2000, 2005 and 2010, respectively, compared to our local estimates of 49.23%, 53.02% and 56.17% for the same corresponding period, showing decreasing differences over time between HK and mainland China schoolchildren from 2000 to 2010 [21].

Overall prevalence of reduced VA was negatively associated with the proportion of primary to secondary schoolchildren. Although subgroup prevalence of each age group remains the same, differences in annual birth rate can cause fluctuation of age composition of study population and influence the overall prevalence, so that a smaller proportion of younger or junior students would raise the overall prevalence. Thus, we used a direct standardization method to adjust the age and sex composition from sampling weight to standardized weight for reviewing the prevalence trend of reduced VA, which was not considered in the two previous studies from mainland China.

### 4.2. Age and Sex

Although univariate and multivariate logistic analysis showed that girls were more susceptible than boys to having reduced VA, our subgroup analysis actually showed that being a girl was less susceptible at age 6−7 or grade P1-P2, and only became more susceptible when older. Significant interaction term showed that there would be multiplicative effect of combining age (or grade) with sex to have a joint effect greater than the product of their individual effects. This finding correlates with the higher incidence of myopic progression among girls compared to boys noted in most studies [3,33,34,35,36,37] and that this tends to occur as they reach adolescence [38]. Possible explanations included changes in lifestyle behavior influencing near vision usage, and the earlier onset of puberty among girls [39]. 

### 4.3. Student Health Service Centres

Since one SHSC served students coming from various school or home districts, we assumed that variations would exist among SHSCs. In the univariate analysis, SHSCs contributed to the variations on the prevalence of reduced VA. After adjusting for other variables, the risk effect of SHSCs reduced but still existed. We expected that the background of the students, for example, age, sex, home or school district, would partially but not fully explain the variations caused by SHSCs. 

### 4.4. School Type and Grade of Students

The prevalence of reduced VA changed disproportionally among different age or grade subgroups. Throughout the study period, the magnitude of fluctuation of prevalence in younger age (6−9 years) or junior grade (P1–P4) was greater than that in older age (10–15 years) or senior grade (P5–S4). The ORs of secondary to primary school, and senior to junior grade, decreased from 2000/2001−2003/2004 to 2013/2014−2016/2017, reflecting a rising prevalence of reduced VA among primary school and junior grade students. Since the change of subgroup prevalence was uneven, more obvious at junior grade and less obvious at senior grade, we believed the onset of reduced VA is occurring earlier, and lessening the effect of schooling duration (or academic attainment) in the senior grade over the past 17 years. 

Large relative changes in ORs of grade (and school type) on reduced VA indicated the presence of the confounding effect of age on influencing the observed association between grade (and school type) and reduced VA in univariate and multivariate analysis. The adjusted measure gave a better correlation and estimate than using the crude measure, since it disregarded the confounding effect of age. Without this adjustment, a negative confounding effect would result in an underestimation of the risk effect of grade during 2000/2001−2003/2004, and inversely, a positive confounding effect would overestimate the risk effect of grade from 2004/2005 to 2016/2017 and the risk effect of school type from 2000/2001 to 2016/2017. The rising relative change in ORs showed that the extent of overestimating the association between grade (and school type) and reduced VA had enlarged. Since the confounding effect reflected the natural relationships between lifestyle, habits and other characteristics, we expect age might possibly be associated with other potential risk factors for reduced VA which are not covered in our study. 

### 4.5. Living Environment and Social Classes

Living in villas or squatter was associated with a lower risk for reduced VA relative to public or private apartment blocks. A possible explanation may be that villas and squatter are usually located in less developed areas with a lower population and building density, while apartment blocks are more likely to be located in urbanized areas with a higher population and building density. This finding was consistent with a previous study showing that a greater number of floors/levels of housings or higher frequency of seeing green were related to increasing prevalence of myopia, one major cause of reduced VA [27,28].

Relative to subsidized HOS flats and public housing (or squatter), living in private housing (whether apartments or villas) was associated with a higher risk for reduced VA. This is similar to previous report of positive association between myopia and socioeconomic status, since only households with higher than average income can afford private housing in HK [26]. Interestingly, the ORs of private housing relative to subsidized HOS flats and public rental housing (or villas relative to squatter) decreased from 2000/2001−2003/2004 to 2013/2014−2016/2017, indicating a lessening of the association between socioeconomic status and reduced VA, possibly from a significant increase in the use of mobile electronic devices among all social classes across HK during this period.

### 4.6. Geographic Variation and Spatial Autocorrelation

Despite the fact that HK has a total area of 1108 km^2^ only, the spatial location was not independent of the presence of reduced VA. There was geographic variation and positive spatial autocorrelation between the spatial location and presence of reduced VA on the map. In 2000/2001−2003/2004, school districts were not spatially clustered while home districts with the lowest AOR were located at the northern areas of HK (Tuen Mun, Yuen Long, North and Tai Po districts), where there is a lower population density and urbanization. During 2013/2014−2016/2017, school districts were also not spatially clustered, while home districts with the highest AOR were located at the southern areas of HK (Central and Western, Wan Chai, Eastern, Southern and Kowloon City districts), where there is a high population density and urbanization. Our findings are consistent with previous studies that showed living environment being a key factor for myopia prevalence, which is a major cause of reduced VA. 

Compared to the student’s home district, most school districts had lower AOR for reduced VA in 2000/2001−2003/2004, but higher AOR by 2013/2014−2016/217, with 15 school districts having lower AOR for reduced VA than home districts in 2000/2001−2003/2004, with only one school district having lower AOR and seven school districts having higher AOR by 2013/2014−2016/2017. Between the periods of 2000/2001−2003/2004 and 2013/2014−2016/2017, 11 home districts had a decreasing trend of AORs and 13 school districts had an increasing trend of AORs. This may be related to the relatively static nature of most schools compared to homes in HK, as increasing urbanization occurs around schools initially established in less urbanized areas, while rising property prices encourage young families with children to move to less expensive and less urbanized areas, further away from the central areas of HK.

There were one school district and 13 home districts with insignificant AORs in 2000/2001−2003/2004, as well as eight school districts and four home districts with insignificant AORs in 2013/2014−2016/2017. AORs tended to be close to 1 and their mean on average by school district, but far from 1 and their mean on average by home district. We expected that the effect of geographic variation on AORs tended to be mitigated by school district but increased by home district. 

The result of measuring spatial autocorrelation differed between school and home districts. Mantel test and Moran’s Index reported higher value of correlation coefficients for home district than school district. Home districts with similar AORs were likely to be clustered together, but this clustering did occur for school districts with similar AORs. Spatial location was significantly correlated to the presence of reduced VA by home district but not school district.

We deduced that the association of spatial location and attributes of the 18 districts contributed to the distinct results of AORs, geographic variation and spatial autocorrelation between school district and home district. The attributes of school districts (for example; culture, academic performance or teaching style) were less likely to be associated with the community distance from city center (located at the Central and Western District), in contrast to the attributes of home districts (for example; population density, urbanization and greenery ratio) which were more likely to be associated. Thus, home districts (but not school districts) with similar attributes were more likely to be clustered together in relation to the community distance from city center. 

### 4.7. Cross-Boundary Students

Mainland China started to be an option of student’s home district from 2013/2014 onwards. The AORs of schoolchildren living in mainland China (but commuting daily to HK to attend school) relative to the Central and Western District in HK were 0.499 and 0.477 in 2013/2014 and 2016/2017, respectively, so that cross-boundary students were less susceptible to reduced VA than local HK students (Appendix A). Moreover, the fertility trend in HK reported that the proportion of live births to non-HK citizen parents increased from 1.29% in 2001 to 29.18% in 2012, when HK became a popular destination for birth tourism for mainland Chinese (Appendix A). It is possible that the decreasing prevalence of reduced VA among the younger age or junior grade students starting from 2012/2013 may be related to the increased proportion of such students who often resides in less densely populated areas across the border from HK. 

Since 2013, birth tourism has decreased greatly after new restrictions on cross-border entry were adopted by the HK Government. As the school entrance age is generally around 6, we postulate that the prevalence of reduced VA among the newly admitted students would increase again from 2019/2020 onwards, to reflect the previous local trend before the large influx of cross-border students. We recommended conducting further study to compare the environmental association on the prevalence of reduced VA between children residing locally in HK with those residing in mainland China (but attending school in HK). 

### 4.8. Strengths and Limitations

The overall prevalence of reduced VA was negatively associated with the proportion of primary to secondary schoolchildren. Assuming that the prevalence of reduced VA in each age or grade subgroup remained the same, fluctuation in the composition of study population would change the overall prevalence, that is, a smaller proportion of younger or junior students would raise the overall prevalence. Thus, we used direct standardization method to adjust the age and sex composition from sampling weight to standardized weight for reviewing the trend of prevalence of reduced VA, which was not considered in the two previous studies from mainland China.

Our study considered interaction effect, confounding, geographic features and dynamic changes over time, which were not reported previously. Significant interaction effect identified that the risk associated with being a girl relative to a boy was increasingly shifted to a later stage of development. Age was the common cause to the grade of students and presence of reduced VA. Geographic variation on the risk for reduced VA existed and spatial autocorrelation was positive. Cross-border students had a lower prevalence of reduced VA than local students, and may have been related to the decreasing prevalence of reduced VA among younger students since 2012/2013.

The limitation was that we considered personal-level factors but did not cover the district-level factors and measure their risk effect in our analysis. We recommended conducting a multilevel research to examine how the compositional and contextual factors affect the prevalence of reduced VA by combining the district-level data from other official departments such as median monthly household income, household size, population density and greenery scale [40,41].

## 5. Conclusions

Since the prevalence of reduced VA was negatively associated with the proportion of primary to secondary schoolchildren, it is important to adjust the sampling weight of age and sex composition to standardized weight in a serial cross-sectional study. In subgroup analysis, the interaction effect between age (or grade) and sex was statistically significant. Girls were less susceptible than boys to reduced VA at age 6−7 or grade P1–P2 but became more susceptible at older ages. There was an increasing prevalence of reduced VA in junior grade so that the onset of reduced VA has been advanced earlier, thus lessening the risk effect of higher grades over the past 17 years. Schoolchildren in HK had lower prevalence of myopia than those in Taiwan, whereas the difference of prevalence of reduced VA between HK and mainland China has lessened with time. The association of housing type with reduced VA is likely affected by both living environment and social classes. Although the area of HK is small, there are geographic variation on the risk for reduced VA and positive spatial autocorrelation. Cross-border students living in mainland China was associated with a lower risk for reduced VA and further study should be considered for investigating the environmental association between students living in and outside HK with the prevalence of reduced VA. We did not include district-level factors in our analysis and suggested to perform a multilevel research to examine how the compositional and contextual factors affect the prevalence of reduced VA.

## Figures and Tables

**Figure 1 ijerph-17-01023-f001:**
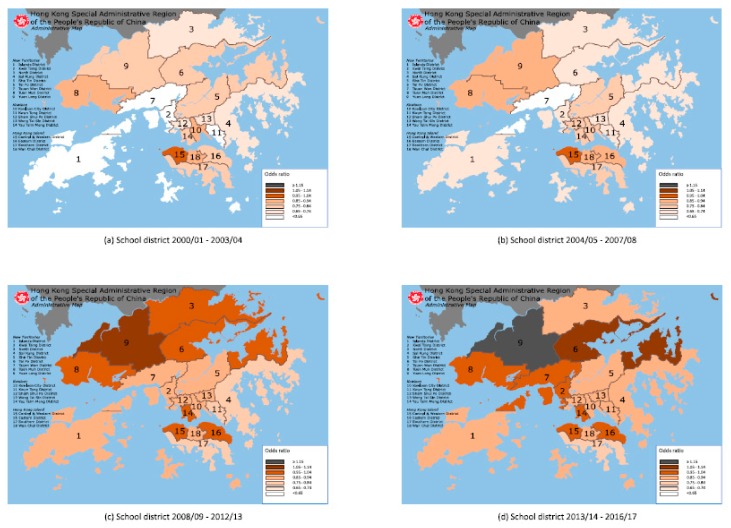
Map of 18 administrative districts in Hong Kong showing the adjusted odds ratio for reduced visual acuity by student’s school district (17 districts relative to Central & Western District) during 2000/01–2003/04, 2004/05–2007/08, 2008/09–2012/13, and 2013/14–2016/17 (Modified map from original source: Wikimedia Commons (https://commons.wikimedia.org/wiki/File:Map_of_Hong_Kong_18_Districts_en.svg))

**Figure 2 ijerph-17-01023-f002:**
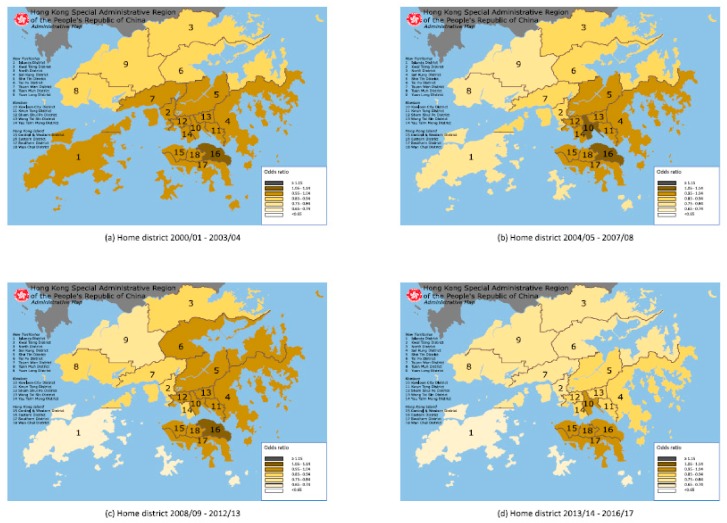
Map of 18 administrative districts in Hong Kong showing the adjusted odds ratio for reduced visual acuity by student’ s home district (17 districts relative to Central & Western District) during 2000/01–2003/04, 2004/05–2007/08, 2008/09–2012/13, and 2013/14–2016/17 (Modified map from original source: Wikimedia Commons (https://commons.wikimedia.org/wiki/File:Map_of_Hong_Kong_18_Districts_en.svg)).

**Table 1 ijerph-17-01023-t001:** Prevalence of reduced visual acuity in the schoolchildren of Hong Kong and its correlation with the proportion of primary to secondary schoolchildren from 2000/2001 to 2016/2017.

School Year	Crude Prevalence Rate			Age-Sex Adjusted Prevalence Rate			Standardized Prevalence Rate			Proportion of Primary to Secondary Schoolchildren	
	Estimate	95% LB	95% UB	Estimate	95% LB	95% UB	Estimate	95% LB	95% UB	Students Attending SHS	Enrolled Students
2000/2001	45.46	45.31	45.61	49.23	49.07	49.39	**49.23**	**48.99**	**49.47**	4.26	1.54
2001/2002	46.03	45.89	46.17	49.42	49.26	49.58	49.45	49.22	49.68	3.88	1.53
2002/2003	45.93	45.78	46.07	49.10	48.95	49.25	48.97	48.74	49.19	3.58	1.5
2003/2004	47.51	47.37	47.65	50.14	50.00	50.29	49.75	49.54	49.97	3.21	1.44
2004/2005	49.41	49.27	49.55	51.87	51.73	52.02	50.98	50.76	51.19	2.98	1.36
2005/2006	51.05	50.91	51.19	53.02	52.87	53.17	51.73	51.51	51.94	2.68	1.29
2006/2007	51.78	51.64	51.92	53.62	53.47	53.77	52.00	51.78	52.21	2.45	1.23
2007/2008	52.86	52.71	53.01	54.51	54.36	54.66	52.67	52.45	52.89	2.26	1.16
2008/2009	53.99	53.84	54.14	55.72	55.56	55.87	53.64	53.42	53.87	2.1	1.1
2009/2010	-	-	-	-	-	-	-	-	-	-	-
2010/2011	54.93	54.78	55.09	56.17	56.01	56.32	54.08	53.86	54.31	1.84	1.06
2011/2012	54.36	54.20	54.52	55.99	55.83	56.15	**54.34**	**54.10**	**54.58**	2.06	1.1
2012/2013	50.98	50.82	51.14	52.81	52.64	52.97	51.59	51.36	51.83	2.1	1.14
2013/2014	49.74	49.58	49.91	51.62	51.46	51.79	51.18	50.94	51.42	2.21	1.22
2014/2015	49.64	49.48	49.80	51.25	51.08	51.41	51.58	51.34	51.82	2.34	1.3
2015/2016	49.35	49.19	49.51	50.89	50.73	51.06	51.70	51.46	51.95	2.53	1.39
2016/2017	48.65	48.49	48.81	50.28	50.11	50.44	**51.42**	**51.17**	**51.66**	2.75	1.49
Mann-Kendall Trend Test											
Time series		tau	*p*-value		tau	*p*-value		tau	*p*-value		
Overall											
2000/2001 – 2016/2017		0.267	0.163		0.217	0.26		**0.383**	**0.043**		
Breakdown											
2000/2001 – 2011/2012		0.927	<0.001		0.891	<0.001		**0.927**	<**0.001**		
2012/2013 – 2016/2017		−1	0.027		−1	0.027		**0**	**1**		
Pearson correlation coefficient (correlation of prevalence of reduced VA with proportion of primary to secondary students attending SHS and enrolled students)											
Students attending SHS			−0.891			−0.809			−**0.875**		
Enrolled students			−0.947			−0.940			−**0.885**		

LB, lower bound; UB, upper bound; SHS, Student Health Service; VA, visual acuity; Bolded figures were referred in Section 3.2; School year 2009/2010 was excluded because SHS limited the annual appointments in that year for taking part in the Human Swine Influenza Vaccination Program.

**Table 2 ijerph-17-01023-t002:** Interaction effect of age (or grade) on the association between sex and reduced visual acuity.

**(a) Effect Modifier: Age**																		
			2000/2001		2001/2002		2002/2003		2003/2004		2004/2005		2005/2006		2006/2007		2007/2008	
			OR	*p*-value	OR	*p*-value	OR	*p*-value	OR	*p*-value	OR	*p*-value	OR	*p*-value	OR	*p*-value	OR	*p*-value
Intercept			0.336	<0.0001	0.326	<0.0001	0.289	<0.0001	0.317	<0.0001	0.305	<0.0001	0.314	<0.0001	0.322	<0.0001	0.329	<0.0001
age	7		**1.163**	<**0.0001**	1.263	<0.0001	1.335	<0.0001	1.241	<0.0001	1.351	<0.0001	1.375	<0.0001	1.352	<0.0001	1.375	<0.0001
age	8		**1.613**	<**0.0001**	1.66	<0.0001	1.813	<0.0001	1.762	<0.0001	1.977	<0.0001	1.941	<0.0001	1.965	<0.0001	1.958	<0.0001
age	9		2.145	<0.0001	2.158	<0.0001	2.475	<0.0001	2.27	<0.0001	2.623	<0.0001	2.632	<0.0001	2.594	<0.0001	2.733	<0.0001
age	10		2.825	<0.0001	2.817	<0.0001	3.079	<0.0001	2.95	<0.0001	3.283	<0.0001	3.342	<0.0001	3.296	<0.0001	3.322	<0.0001
age	11		3.376	<0.0001	3.586	<0.0001	3.938	<0.0001	3.598	<0.0001	4.023	<0.0001	4.112	<0.0001	4.052	<0.0001	4.143	<0.0001
age	12		4.054	<0.0001	4.234	<0.0001	4.861	<0.0001	4.42	<0.0001	4.738	<0.0001	4.833	<0.0001	4.721	<0.0001	4.819	<0.0001
age	13		4.652	<0.0001	4.849	<0.0001	5.509	<0.0001	5.319	<0.0001	5.569	<0.0001	5.364	<0.0001	5.533	<0.0001	5.469	<0.0001
age	14		5.264	<0.0001	5.568	<0.0001	6.228	<0.0001	5.771	<0.0001	6.59	<0.0001	6.207	<0.0001	5.867	<0.0001	5.976	<0.0001
age	15		**5.525**	<**0.0001**	5.731	<0.0001	7.068	<0.0001	6.641	<0.0001	6.949	<0.0001	7.001	<0.0001	6.635	<0.0001	6.451	<0.0001
sex	F		**0.893**	<**0.0001**	0.96	<0.0001	0.958	<0.0001	0.912	<0.0001	0.943	<0.0001	0.946	<0.0001	0.913	<0.0001	0.946	<0.0001
age*sex	7	F	**1.095**	<**0.0001**	0.966	<0.0001	0.985	0.0278	1.062	<0.0001	1.033	<0.0001	1.029	0.0001	1.04	<0.0001	1.038	<0.0001
age*sex	8	F	**1.151**	<**0.0001**	1.057	<0.0001	1.042	<0.0001	1.081	<0.0001	1.056	<0.0001	1.094	<0.0001	1.11	<0.0001	1.05	<0.0001
age*sex	9	F	1.164	<0.0001	1.142	<0.0001	1.074	<0.0001	1.191	<0.0001	1.102	<0.0001	1.139	<0.0001	1.216	<0.0001	1.112	<0.0001
age*sex	10	F	1.207	<0.0001	1.149	<0.0001	1.186	<0.0001	1.214	<0.0001	1.191	<0.0001	1.186	<0.0001	1.264	<0.0001	1.227	<0.0001
age*sex	11	F	1.314	<0.0001	1.186	<0.0001	1.2	<0.0001	1.339	<0.0001	1.287	<0.0001	1.256	<0.0001	1.291	<0.0001	1.269	<0.0001
age*sex	12	F	1.337	<0.0001	1.246	<0.0001	1.195	<0.0001	1.308	<0.0001	1.332	<0.0001	1.309	<0.0001	1.357	<0.0001	1.264	<0.0001
age*sex	13	F	1.367	<0.0001	1.272	<0.0001	1.241	<0.0001	1.247	<0.0001	1.275	<0.0001	1.348	<0.0001	1.331	<0.0001	1.28	<0.0001
age*sex	14	F	1.274	<0.0001	1.24	<0.0001	1.215	<0.0001	1.334	<0.0001	1.194	<0.0001	1.235	<0.0001	1.333	<0.0001	1.279	<0.0001
age*sex	15	F	**1.361**	<**0.0001**	1.275	<0.0001	1.214	<0.0001	1.205	<0.0001	1.256	<0.0001	1.186	<0.0001	1.297	<0.0001	1.257	<0.0001
			2008/2009		2010/2011		2011/2012		2012/2013		2013/2014		2014/2015		2015/2016		2016/2017	
			OR	*p*-value	OR	*p*-value	OR	*p*-value	OR	*p*-value	OR	*p*-value	OR	*p*-value	OR	*p*-value	OR	*p*-value
Intercept			0.36	<0.0001	0.376	<0.0001	0.404	<0.0001	0.252	<0.0001	0.256	<0.0001	0.264	<0.0001	0.275	<0.0001	0.268	<0.0001
age	7		1.349	<0.0001	1.367	<0.0001	1.341	<0.0001	1.662	<0.0001	1.535	<0.0001	1.529	<0.0001	1.45	<0.0001	1.504	<0.0001
age	8		1.883	<0.0001	1.978	<0.0001	1.898	<0.0001	2.702	<0.0001	2.46	<0.0001	2.356	<0.0001	2.23	<0.0001	2.165	<0.0001
age	9		2.539	<0.0001	2.563	<0.0001	2.459	<0.0001	3.685	<0.0001	3.495	<0.0001	3.403	<0.0001	3.094	<0.0001	3.057	<0.0001
age	10		3.199	<0.0001	3.129	<0.0001	3.001	<0.0001	4.492	<0.0001	4.502	<0.0001	4.314	<0.0001	4.134	<0.0001	3.91	<0.0001
age	11		3.939	<0.0001	3.838	<0.0001	3.566	<0.0001	5.392	<0.0001	5.322	<0.0001	5.33	<0.0001	5.179	<0.0001	5.079	<0.0001
age	12		4.677	<0.0001	4.439	<0.0001	4.076	<0.0001	6.257	<0.0001	6.185	<0.0001	6.14	<0.0001	5.852	<0.0001	5.887	<0.0001
age	13		5.196	<0.0001	4.927	<0.0001	4.599	<0.0001	6.818	<0.0001	6.761	<0.0001	6.466	<0.0001	6.432	<0.0001	6.791	<0.0001
age	14		5.623	<0.0001	5.41	<0.0001	4.886	<0.0001	7.534	<0.0001	7.261	<0.0001	7.201	<0.0001	6.941	<0.0001	7.103	<0.0001
age	15		6.215	<0.0001	5.932	<0.0001	5.365	<0.0001	8.042	<0.0001	8.194	<0.0001	7.489	<0.0001	7.419	<0.0001	7.616	<0.0001
sex	F		0.917	<0.0001	0.91	<0.0001	0.916	<0.0001	0.875	<0.0001	0.897	<0.0001	0.868	<0.0001	0.931	<0.0001	0.953	<0.0001
age*sex	7	F	1.036	<0.0001	1.036	<0.0001	0.996	0.5881	1.038	<0.0001	1.023	0.0035	1.102	<0.0001	1.022	0.0052	0.97	<0.0001
age*sex	8	F	1.113	<0.0001	1.065	<0.0001	1.037	<0.0001	1.074	<0.0001	1.086	<0.0001	1.109	<0.0001	1.045	<0.0001	1.058	<0.0001
age*sex	9	F	1.145	<0.0001	1.161	<0.0001	1.155	<0.0001	1.124	<0.0001	1.119	<0.0001	1.188	<0.0001	1.122	<0.0001	1.079	<0.0001
age*sex	10	F	1.235	<0.0001	1.236	<0.0001	1.19	<0.0001	1.256	<0.0001	1.135	<0.0001	1.238	<0.0001	1.17	<0.0001	1.184	<0.0001
age*sex	11	F	1.303	<0.0001	1.239	<0.0001	1.246	<0.0001	1.306	<0.0001	1.254	<0.0001	1.255	<0.0001	1.225	<0.0001	1.226	<0.0001
age*sex	12	F	1.309	<0.0001	1.308	<0.0001	1.241	<0.0001	1.31	<0.0001	1.285	<0.0001	1.34	<0.0001	1.242	<0.0001	1.29	<0.0001
age*sex	13	F	1.298	<0.0001	1.33	<0.0001	1.264	<0.0001	1.346	<0.0001	1.304	<0.0001	1.417	<0.0001	1.321	<0.0001	1.299	<0.0001
age*sex	14	F	1.287	<0.0001	1.299	<0.0001	1.336	<0.0001	1.335	<0.0001	1.296	<0.0001	1.363	<0.0001	1.327	<0.0001	1.323	<0.0001
age*sex	15	F	1.289	<0.0001	1.24	<0.0001	1.322	<0.0001	1.427	<0.0001	1.273	<0.0001	1.462	<0.0001	1.339	<0.0001	1.346	<0.0001
**(b) Effect Modifier: Grade**																		
			2000/2001		2001/2002		2002/2003		2003/2004		2004/2005		2005/2006		2006/2007		2007/2008	
			OR	*p*-value	OR	*p*-value	OR	*p*-value	OR	*p*-value	OR	*p*-value	OR	*p*-value	OR	*p*-value	OR	*p*-value
Intercept			0.331	<0.0001	0.333	<0.0001	0.297	<0.0001	0.321	<0.0001	0.311	<0.0001	0.333	<0.0001	0.335	<0.0001	0.349	<0.0001
GRADE	P2		**1.292**	<**0.0001**	1.343	<0.0001	1.45	<0.0001	1.4	<0.0001	1.561	<0.0001	1.468	<0.0001	1.503	<0.0001	1.486	<0.0001
GRADE	P3		**1.805**	<**0.0001**	1.763	<0.0001	1.955	<0.0001	1.893	<0.0001	2.141	<0.0001	2.051	<0.0001	2.086	<0.0001	2.114	<0.0001
GRADE	P4		2.437	<0.0001	2.351	<0.0001	2.622	<0.0001	2.491	<0.0001	2.81	<0.0001	2.777	<0.0001	2.776	<0.0001	2.821	<0.0001
GRADE	P5		3.148	<0.0001	3.063	<0.0001	3.345	<0.0001	3.203	<0.0001	3.483	<0.0001	3.462	<0.0001	3.514	<0.0001	3.443	<0.0001
GRADE	P6		3.896	<0.0001	3.982	<0.0001	4.298	<0.0001	3.994	<0.0001	4.32	<0.0001	4.208	<0.0001	4.259	<0.0001	4.308	<0.0001
GRADE	S1		5.01	<0.0001	4.733	<0.0001	5.411	<0.0001	4.937	<0.0001	5.159	<0.0001	4.931	<0.0001	4.994	<0.0001	4.949	<0.0001
GRADE	S2		5.596	<0.0001	5.633	<0.0001	6.14	<0.0001	5.882	<0.0001	6.116	<0.0001	5.477	<0.0001	5.56	<0.0001	5.54	<0.0001
GRADE	S3		6.198	<0.0001	6.525	<0.0001	7.175	<0.0001	6.477	<0.0001	7.136	<0.0001	6.382	<0.0001	6.296	<0.0001	5.995	<0.0001
GRADE	S4		**6.827**	<**0.0001**	6.578	<0.0001	8.182	<0.0001	7.408	<0.0001	7.293	<0.0001	7.417	<0.0001	6.874	<0.0001	6.565	<0.0001
sex	F		**0.909**	<**0.0001**	0.929	<0.0001	0.938	<0.0001	0.922	<0.0001	0.945	<0.0001	0.944	<0.0001	0.934	<0.0001	0.958	<0.0001
GRADE*sex	P2	F	**1.085**	<**0.0001**	1.038	<0.0001	1.03	<0.0001	1.06	<0.0001	1.032	<0.0001	1.029	<0.0001	1.028	<0.0001	1.008	0.2174
GRADE*sex	P3	F	**1.137**	<**0.0001**	1.114	<0.0001	1.066	<0.0001	1.105	<0.0001	1.082	<0.0001	1.136	<0.0001	1.102	<0.0001	1.045	<0.0001
GRADE*sex	P4	F	1.165	<0.0001	1.179	<0.0001	1.127	<0.0001	1.17	<0.0001	1.116	<0.0001	1.142	<0.0001	1.223	<0.0001	1.146	<0.0001
GRADE*sex	P5	F	1.235	<0.0001	1.218	<0.0001	1.231	<0.0001	1.237	<0.0001	1.223	<0.0001	1.206	<0.0001	1.241	<0.0001	1.243	<0.0001
GRADE*sex	P6	F	1.316	<0.0001	1.237	<0.0001	1.218	<0.0001	1.294	<0.0001	1.29	<0.0001	1.274	<0.0001	1.279	<0.0001	1.249	<0.0001
GRADE*sex	S1	F	1.302	<0.0001	1.324	<0.0001	1.244	<0.0001	1.292	<0.0001	1.309	<0.0001	1.331	<0.0001	1.303	<0.0001	1.232	<0.0001
GRADE*sex	S2	F	1.293	<0.0001	1.267	<0.0001	1.244	<0.0001	1.26	<0.0001	1.238	<0.0001	1.32	<0.0001	1.32	<0.0001	1.233	<0.0001
GRADE*sex	S3	F	1.198	<0.0001	1.263	<0.0001	1.197	<0.0001	1.235	<0.0001	1.186	<0.0001	1.203	<0.0001	1.255	<0.0001	1.256	<0.0001
GRADE*sex	S4	F	**1.273**	<**0.0001**	1.197	<0.0001	1.183	<0.0001	1.177	<0.0001	1.254	<0.0001	1.117	<0.0001	1.256	<0.0001	1.231	<0.0001
			2008/2009		2010/2011		2011/2012		2012/2013		2013/2014		2014/2015		2015/2016		2016/2017	
			OR	*p*-value	OR	*p*-value	OR	*p*-value	OR	*p*-value	OR	*p*-value	OR	*p*-value	OR	*p*-value	OR	*p*-value
Intercept			0.38	<0.0001	0.399	<0.0001	0.414	<0.0001	0.272	<0.0001	0.275	<0.0001	0.284	<0.0001	0.291	<0.0001	0.291	<0.0001
GRADE	P2		1.441	<0.0001	1.459	<0.0001	1.503	<0.0001	1.827	<0.0001	1.666	<0.0001	1.695	<0.0001	1.614	<0.0001	1.574	<0.0001
GRADE	P3		1.964	<0.0001	2.041	<0.0001	2.034	<0.0001	2.781	<0.0001	2.616	<0.0001	2.485	<0.0001	2.362	<0.0001	2.253	<0.0001
GRADE	P4		2.71	<0.0001	2.647	<0.0001	2.667	<0.0001	3.748	<0.0001	3.618	<0.0001	3.538	<0.0001	3.308	<0.0001	3.182	<0.0001
GRADE	P5		3.248	<0.0001	3.179	<0.0001	3.093	<0.0001	4.444	<0.0001	4.489	<0.0001	4.345	<0.0001	4.266	<0.0001	3.972	<0.0001
GRADE	P6		4.094	<0.0001	3.956	<0.0001	3.793	<0.0001	5.394	<0.0001	5.388	<0.0001	5.498	<0.0001	5.271	<0.0001	5.164	<0.0001
GRADE	S1		4.749	<0.0001	4.342	<0.0001	4.132	<0.0001	5.961	<0.0001	5.9	<0.0001	5.742	<0.0001	5.662	<0.0001	5.641	<0.0001
GRADE	S2		5.243	<0.0001	4.989	<0.0001	4.741	<0.0001	6.716	<0.0001	6.568	<0.0001	6.376	<0.0001	6.461	<0.0001	6.604	<0.0001
GRADE	S3		5.883	<0.0001	5.254	<0.0001	4.88	<0.0001	7.235	<0.0001	7.019	<0.0001	6.762	<0.0001	6.644	<0.0001	6.673	<0.0001
GRADE	S4		6.103	<0.0001	6.004	<0.0001	5.913	<0.0001	8.1	<0.0001	8.354	<0.0001	7.45	<0.0001	7.502	<0.0001	7.346	<0.0001
sex	F		0.918	<0.0001	0.903	<0.0001	0.916	<0.0001	0.875	<0.0001	0.903	<0.0001	0.889	<0.0001	0.937	<0.0001	0.919	<0.0001
GRADE*sex	P2	F	1.057	<0.0001	1.058	<0.0001	1.02	0.0032	1.051	<0.0001	1.015	0.0358	1.065	<0.0001	0.998	0.7694	1.073	<0.0001
GRADE*sex	P3	F	1.117	<0.0001	1.103	<0.0001	1.067	<0.0001	1.115	<0.0001	1.073	<0.0001	1.093	<0.0001	1.069	<0.0001	1.083	<0.0001
GRADE*sex	P4	F	1.148	<0.0001	1.185	<0.0001	1.135	<0.0001	1.149	<0.0001	1.147	<0.0001	1.159	<0.0001	1.114	<0.0001	1.135	<0.0001
GRADE*sex	P5	F	1.264	<0.0001	1.23	<0.0001	1.224	<0.0001	1.259	<0.0001	1.136	<0.0001	1.247	<0.0001	1.172	<0.0001	1.241	<0.0001
GRADE*sex	P6	F	1.318	<0.0001	1.276	<0.0001	1.23	<0.0001	1.313	<0.0001	1.253	<0.0001	1.202	<0.0001	1.238	<0.0001	1.274	<0.0001
GRADE*sex	S1	F	1.316	<0.0001	1.326	<0.0001	1.258	<0.0001	1.334	<0.0001	1.32	<0.0001	1.38	<0.0001	1.27	<0.0001	1.394	<0.0001
GRADE*sex	S2	F	1.241	<0.0001	1.357	<0.0001	1.279	<0.0001	1.316	<0.0001	1.271	<0.0001	1.359	<0.0001	1.282	<0.0001	1.295	<0.0001
GRADE*sex	S3	F	1.246	<0.0001	1.281	<0.0001	1.374	<0.0001	1.345	<0.0001	1.273	<0.0001	1.35	<0.0001	1.323	<0.0001	1.39	<0.0001
GRADE*sex	S4	F	1.325	<0.0001	1.204	<0.0001	1.248	<0.0001	1.42	<0.0001	1.228	<0.0001	1.376	<0.0001	1.267	<0.0001	1.38	<0.0001

Bolded figures were referred in Section 3.3; age*sex, interaction term between age and sex; GRADE*sex, interaction term between grade and sex; School year 2009/2010 was excluded because SHS limited the annual appointments in that year for taking part in the Human Swine Influenza Vaccination Program.

**Table 3 ijerph-17-01023-t003:** Factors associated with reduced visual acuity in the schoolchildren of Hong Kong during 2000/2001−2003/2004, 2004/2005−2007/2008, 2008/2009−2012/2013 and 2013/2014−2016/2017: Multivariate logistic regression.

		2000/2001–2003/2004			2004/2005–2007/2008			2008/2009–2012/2013			2013/2014–2016/2017		
		OR	95% LB	95% UB	OR	95% LB	95% UB	OR	95% LB	95% UB	OR	95% LB	95% UB
age		**0.974**	**0.969**	**0.979**	1.035	1.030	1.040	1.062	1.056	1.068	**1.097**	**1.091**	**1.104**
Sex	Boys (ref)												
	Girls	1.102	1.095	1.109	1.114	1.108	1.121	1.078	1.071	1.085	1.087	1.079	1.094
School District	Central and Western (ref)												
	Wan Chai	0.909	0.884	0.934	0.924	0.898	0.951	0.932	0.904	0.960	0.939	0.910	0.968
	Eastern	0.847	0.816	0.880	0.911	0.878	0.944	0.992	0.955	1.030	1.008	0.969	1.049
	Southern	0.848	0.824	0.874	0.823	0.798	0.847	0.826	0.802	0.852	0.825	0.799	0.851
	Yau Tsim Mong	0.824	0.765	0.886	0.841	0.798	0.886	0.963	0.916	1.012	1.006	0.958	1.057
	Sham Shui Po	0.737	0.684	0.794	0.790	0.749	0.833	0.896	0.853	0.941	0.929	0.885	0.976
	Kowloon City	0.865	0.804	0.931	0.848	0.805	0.893	0.916	0.872	0.962	0.942	0.898	0.989
	Wong Tai Sin	0.678	0.629	0.731	0.712	0.675	0.752	0.896	0.851	0.943	0.933	0.887	0.983
	Kwun Tong	0.588	0.546	0.633	0.656	0.623	0.692	0.768	0.731	0.807	0.877	0.835	0.921
	Tsuen Wan	0.576	0.538	0.618	0.639	0.603	0.676	0.893	0.844	0.945	1.036	0.978	1.098
	Tuen Mun	0.867	0.733	1.027	0.911	0.803	1.032	1.022	0.903	1.156	1.021	0.903	1.154
	Yuen Long	0.805	0.698	0.928	0.933	0.835	1.042	1.149	1.038	1.272	1.255	1.130	1.394
	North	0.664	0.598	0.736	0.738	0.667	0.816	0.969	0.850	1.105	0.893	0.782	1.021
	Tai Po	0.765	0.696	0.842	0.661	0.607	0.719	1.049	0.951	1.156	1.106	0.998	1.226
	Sai Kung	0.650	0.602	0.701	0.676	0.640	0.714	0.840	0.798	0.885	0.926	0.880	0.975
	Sha Tin	0.770	0.707	0.839	0.785	0.735	0.838	0.939	0.881	1.001	0.943	0.884	1.007
	Kwai Tsing	0.616	0.575	0.659	0.673	0.637	0.711	0.877	0.831	0.926	0.970	0.917	1.026
	Islands	0.609	0.574	0.645	0.742	0.696	0.792	0.912	0.861	0.966	0.887	0.835	0.942
Grade of Student	P1 (ref)												
	P2	1.172	1.149	1.196	1.161	1.134	1.188	1.230	1.201	1.260	1.293	1.263	1.324
	P3	1.724	1.689	1.761	1.648	1.608	1.688	1.676	1.634	1.719	1.770	1.724	1.816
	P4	2.377	2.319	2.438	2.150	2.091	2.211	2.138	2.077	2.201	2.310	2.242	2.379
	P5	3.303	3.212	3.397	2.690	2.609	2.774	2.493	2.414	2.575	2.700	2.610	2.793
	P6	4.296	4.159	4.437	3.225	3.114	3.340	2.916	2.810	3.026	3.095	2.976	3.219
	S1	5.550	5.352	5.756	3.704	3.562	3.851	3.094	2.968	3.225	3.153	3.017	3.296
	S2	6.429	6.170	6.698	4.008	3.837	4.186	3.254	3.105	3.410	3.175	3.019	3.338
	S3	7.390	7.058	7.737	4.341	4.137	4.554	3.333	3.166	3.509	3.088	2.922	3.265
	S4	**8.109**	**7.701**	**8.539**	4.621	4.384	4.871	3.589	3.395	3.795	**3.268**	**3.075**	**3.472**
Student Health Service Centre	Chai Wan (ref)												
	Kowloon Bay	1.193	1.106	1.287	1.149	1.090	1.210	1.000	0.952	1.050	1.033	0.984	1.084
	Kowloon City LC	1.143	1.059	1.233	1.107	1.050	1.168	0.987	0.938	1.039	1.056	1.004	1.110
	Lam Tin	1.307	1.211	1.410	1.232	1.167	1.300	1.170	1.112	1.231	1.179	1.122	1.239
	South Kwai Chung	1.410	1.316	1.511	1.340	1.274	1.410	1.168	1.109	1.230	1.145	1.084	1.209
	Sha Tin	1.083	0.993	1.181	1.056	0.987	1.129	0.894	0.837	0.955	0.963	0.900	1.030
	Tai Po	0.989	0.902	1.085	1.153	1.061	1.254	0.731	0.660	0.810	0.886	0.795	0.986
	Shek Wu Hui	1.251	1.129	1.386	1.097	0.991	1.214	0.842	0.736	0.963	1.149	1.002	1.318
	Tuen Mun	0.941	0.797	1.112	0.931	0.818	1.058	0.801	0.705	0.910	0.898	0.790	1.020
	Western	1.093	1.059	1.129	1.058	1.029	1.088	0.998	0.969	1.028	1.001	0.970	1.034
	TWS Wu York Yu	1.130	1.046	1.221	1.058	1.002	1.117	0.959	0.911	1.010	1.107	1.052	1.165
	Yuen Long	1.050	0.911	1.210	1.115	0.995	1.250	0.925	0.832	1.029	0.796	0.713	0.888
Type of Housing	Public rental housing (ref)												
	Subsidized home ownership flats	**1.153**	**1.141**	**1.166**	1.170	1.159	1.183	1.104	1.091	1.117	**1.067**	**1.053**	**1.082**
	Private housing	**1.203**	**1.193**	**1.214**	1.190	1.181	1.200	1.044	1.035	1.053	**0.984**	**0.975**	**0.993**
	Villas/Bungalows/Modern Village Houses	**0.792**	**0.775**	**0.809**	0.786	0.772	0.801	0.781	0.767	0.796	**0.776**	**0.761**	**0.791**
	Squatter/Temp. Housing Area/Stone Hut	**0.607**	**0.589**	**0.626**	0.684	0.661	0.709	0.716	0.684	0.750	**0.745**	**0.702**	**0.790**
Home District	Central and Western (ref)												
	Wan Chai	1.031	0.987	1.076	0.967	0.932	1.004	0.967	0.929	1.006	0.970	0.931	1.012
	Eastern	1.084	1.050	1.118	1.086	1.054	1.119	1.071	1.038	1.105	1.022	0.989	1.057
	Southern	1.016	0.987	1.046	0.990	0.962	1.019	1.008	0.978	1.039	1.020	0.987	1.055
	Yau Tsim Mong	1.002	0.956	1.051	0.981	0.941	1.023	0.940	0.903	0.978	0.885	0.850	0.921
	Sham Shui Po	1.004	0.957	1.054	0.951	0.912	0.992	0.996	0.957	1.037	0.912	0.876	0.949
	Kowloon City	1.026	0.979	1.075	1.052	1.010	1.096	1.042	1.003	1.083	0.996	0.958	1.035
	Wong Tai Sin	1.037	0.988	1.087	1.003	0.962	1.045	0.977	0.938	1.018	0.935	0.898	0.974
	Kwun Tong	1.022	0.976	1.070	0.993	0.954	1.033	0.975	0.939	1.013	0.879	0.846	0.913
	Tsuen Wan	1.008	0.959	1.060	0.945	0.904	0.988	0.880	0.843	0.918	0.839	0.803	0.876
	Tuen Mun	0.866	0.805	0.931	0.825	0.775	0.877	0.884	0.833	0.937	0.842	0.797	0.889
	Yuen Long	0.876	0.818	0.939	0.778	0.734	0.825	0.825	0.781	0.871	0.814	0.774	0.856
	North	0.941	0.884	1.000	0.903	0.854	0.954	0.917	0.872	0.965	0.797	0.759	0.838
	Tai Po	0.899	0.847	0.954	0.934	0.885	0.987	0.965	0.915	1.018	0.775	0.736	0.816
	Sai Kung	1.041	0.992	1.092	1.013	0.972	1.056	0.958	0.921	0.997	0.902	0.868	0.938
	Sha Tin	1.031	0.981	1.084	0.995	0.953	1.040	0.976	0.936	1.018	0.924	0.887	0.964
	Kwai Tsing	0.993	0.945	1.043	0.915	0.876	0.955	0.850	0.816	0.886	0.812	0.779	0.846
	Islands	0.991	0.934	1.052	0.840	0.793	0.889	0.739	0.705	0.775	0.712	0.678	0.747

LB, lower bound; UB, upper bound; SHS, Student Health Service; Bolded figures were referred in Section 3.3, 3.4 and 3.5; School year 2009/2010 was excluded because SHS limited the annual appointments in that year for taking part in the Human Swine Influenza Vaccination Program.

**Table 4 ijerph-17-01023-t004:** Relative change in odds ratio of school type (and grade) on reduced visual acuity before and after adjusting for age in logistic analysis.

	(a) Univariate				(b) Multivariate			
	Relative change in odds ratio (%)				Relative change in odds ratio (%)			
	2000/2001−2003/2004	2004/2005−2007/2008	2008/2009−2012/2013	2012/2013−2016/2017	2000/2001−2003/2004	2004/2005−2007/2008	2008/2009−2012/2013	2012/2013−2016/2017
(i) School type								
Primary School (ref)								
Secondary School	**162.03**	213.89	222.92	**281.33**	**117.48**	131.72	137.09	**191.47**
(ii) Grade								
P1 (ref)								
P2	−4.95	1.93	5.76	10.08	−3.92	4.13	6.83	11.52
P3	−9.62	3.81	11.63	20.94	−6.38	7.89	13.84	22.82
P4	−14.18	5.73	17.93	32.64	−9.3	11.58	21.19	34.85
P5	−18.48	7.74	24.52	45.48	−11.9	15.65	28.92	48.3
P6	−22.43	9.66	31.37	59.16	−14.39	19.75	37.07	62.88
S1	−25.97	11.76	38.84	74.97	−16.54	24.24	46.25	79.67
S2	−29.1	13.69	46.34	91.76	−18.59	28.42	54.98	96.76
S3	−31.8	15.5	53.44	108.65	−20.38	32.14	63.4	114.67
S4	**−33.49**	16.66	58.19	**120.22**	**−21.52**	34.93	68.4	**126.47**

Bolded figures were referred in Section 3.4; School year 2009/2010 was excluded because SHS limited the annual appointments in that year for taking part in the Human Swine Influenza Vaccination Program.

**Table 5 ijerph-17-01023-t005:** Correlation coefficients of Mantel test and Moran’s Index by home district and school district.

		2000/2001−2003/2004		2004/2005−2007/2008		2008/2009−2012/2013		2013/2014−2016/2017	
		Estimate	*p*-Value	Estimate	*p*-Value	Estimate	*p*-Value	Estimate	*p*-Value
(a) Age-Sex Adjusted Prevalence									
(i) Home district									
Mantel Test		0.196	0.077	0.249	0.054	**0.341**	**0.035**	**0.426**	**0.008**
Moran’s Index		0.067	0.463	0.048	0.529	0.103	0.322	0.214	0.107
(ii) School district									
Mantel Test		0.143	0.113	0.099	0.201	0.249	0.074	**0.406**	**0.007**
Moran’s Index		0.053	0.518	−0.02	0.824	0.018	0.644	0.13	0.27
(b) Adjusted Odds Ratio in Multivariate Logistic Analysis									
(i) Home district									
Mantel Test		**0.385**	**0.022**	**0.683**	<**0.001**	**0.636**	<**0.001**	**0.285**	**0.024**
Moran’s Index		0.132	0.249	**0.368**	**0.01**	**0.336**	**0.016**	0.084	0.406
(ii) School district									
Mantel Test		−0.028	0.522	−0.042	0.604	0.113	0.227	0.083	0.249
Moran’s Index		−0.104	0.793	−0.14	0.641	−0.182	0.451	−0.13	0.641

Bolded figures were referred in Section 3.6; School year 2009/2010 was excluded because SHS limited the annual appointments in that year for taking part in the Human Swine Influenza Vaccination Program.

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
