# Peer review of "A Serial Cross-Sectional Analysis of the Prevalence, Risk Factors and Geographic Variations of Reduced Visual Acuity in Primary and Secondary Students from 2000 to 2017 in Hong Kong"

_ijerph, 2020, doi:10.3390/ijerph17031023_

Round 1

Reviewer 1 Report

General Comments

This study covers an important topic, trends in reduced visual acuity (VA) in students in Hong Kong. By combining several data sources, and carefully standardising the main dataset over time, it provides useful data on reduced VA prevalence in Hong Kong across 16 years. It is a clearly written paper, which is mostly easy to follow, logical, well-structured and appropriately referenced. However, more work, especially in presenting and interpreting the results, is needed to prepare the paper for publication and ultimately ensure that it is a valuable addition to the literature.

Introduction

The introduction would benefit from a bit more focus on existing Hong Kong data, and how this compares to data from mainland China/Taiwan. The introduction mentions the limitations of existing Hong Kong data, but not what is currently known (while acknowledging those limitations). Given that results are compared with mainland China/Taiwan in the abstract/conclusion, it might be useful to introduce some ideas about why differences in Hong Kong might be expected. Furthermore, given that trends were investigated (and found) in the data, the fact that existing data from China (and elsewhere) was collected at different time points should be acknowledged. Authors could also mention the importance of standardisation in the introduction, given that your results show the need to standardise carefully. The second paragraph on testing VA and refractive errors initially seemed fairly clear, but actually ultimately I found it quite confusing. I wonder what is the purpose of this section (to make the case for the measure of VA?) and what does this add to the paper. It doesn’t seem to fit the narrative very well. (Perhaps I only say this because I am not an expert on visual health.)

Methods and Results

One very important piece of information is missing from the methods/results: how many students are covered by the annual health check, and what is this as a percentage of Hong Kong students? The results are based on prevalence, but raw descriptive statistics are missing. (Or perhaps I have missed this, but if so, they should be much more prominent.) Before explaining the use of multiple imputation to handle missing data the researchers should say something about how much data (and which data) is actually missing. I understand that for the purpose of presenting the results, choosing 4 time intervals and 5 age/grade intervals may be helpful, but it would be reassuring to know that these choices do not substantially affect the results. The results for grade seem to be basically the same as for age (which is hardly surprising, I assume they correlate very highly). Unless I am missing a good reason to keep both, I think the paper would be easier to read if the authors just state once that results for age and grade were very similar and then subsequently only write about age, with all further references/results regarding grade only included in supplementary files. On a related note, I generally find it useful to report/read correlations between predictors and would encourage the authors to add a table with this information. It took me some time to understand the first paragraph of the results (in particular what the correlation shows, and why it is so strong). I assume what you are trying to say here is that reduced VA prevalence has a very strong negative correlation with SHS attendance/enrolled students because age is a strong predictor of VA? This result might be easier for the reader to understand and interpret if you also explain the crude and age-sex adjusted prevalence rates, and note the differences with the standardised prevalence rate. (Good standardisation practice could also be further discussed in the discussion.) Furthermore, it may be useful to provide readers with some explanation of why the proportion of primary to secondary students attending SHS (and enrolled) has changed so much over time. The tables are difficult to read with the present formatting (so I hope authors/editors will ensure that numbers appear on one line wherever possible, e.g., by setting the relevant pages to ‘landscape’ format.) Table 2 moves back and forth between prevalence data (a/c) and odds ratios (b/d) which is difficult to read. If the researchers accept my suggestion to remove grade data, then c/d are not needed, but if they reject the suggestion, then I think this table should be reordered (e.g., acbd). Prevalence changes over time are one of the main results of the paper, yet I cannot see any statistical testing of whether these changes are significant. Perhaps it is implied by non-overlapping confidence intervals, but the results would be more robust if changes over time were explicitly tested and reported on. To make the odds ratio for age easier to interpret, authors could add ‘per year’ (or describe this in text). The statement “implying that the interaction effect between age and sex was statistically significant” seems very odd, shouldn’t authors just actually test with the interaction effect was significant? (This interaction is also mentioned in the discussion, but I did not see the test result anywhere.) The results regarding geographical variations (Type of housing, School district, Home district and Student Health Service Centres) are inadequately presented and interpreted. Either significant revision is needed to make these results interpretable to people who do not have much knowledge of Hong Kong geography, or these results should be removed (and potentially made the subject of a separate paper). I suggest removal, because these geographical variations are barely mentioned in the introduction, and in the discussion section the explanations are very speculative and completely unreferenced. (Authors even acknowledge this for one of the predictors mentioned: “Due to the limitation of our existing data [on SHSCs], we could not conduct in-depth investigation and provide further explanation.”) I believe that the paper is still sufficient without these results, there is still important data on time trends, age and gender.

Discussion

The discussion needs a lot of work, both on structure and content. Authors should start by clearly laying out their general findings, and only then explaining what these findings add to the existing literature, and how they compare with mainland China/Taiwan etc. Authors should include a section on strengths and limitations. New terms such as myopia are added to the discussion, which also doesn’t help with readability. There is insufficient explanation and interpretation of the interesting findings regarding changes in the effects of gender and age on VA (and the interaction). Possible explanations should be referenced and compared with existing literature. The use of ‘educational attainment’ to refer to year of schooling needs amending or explaining – are these results not just age effects (see also comment 8)? Or are they really about educational attainment? Authors should be very wary of introducing new results (on cross-boundary students) in the discussion. I can see why they may be relevant to the overall trends found in the paper, but I think this idea should be included in the introduction (and thus justifying the including of relevant results in the results section and subsequent discussion). The conclusion should not introduce new ideas (such as ‘the severity of refractive errors’).

Author Response

Reviewer 1

Comment 1:

This study covers an important topic, trends in reduced visual acuity (VA) in students in Hong Kong. By combining several data sources, and carefully standardising the main dataset over time, it provides useful data on reduced VA prevalence in Hong Kong across 16 years.

Authors’ reply:

Yes, we agree, especially no there have been no recent analysis of this issue in our locality.

Comment 2:

It is a clearly written paper, which is mostly easy to follow, logical, well-structured and appropriately referenced. However, more work, especially in presenting and interpreting the results, is needed to prepare the paper for publication and ultimately ensure that it is a valuable addition to the literature.

Authors’ reply:

Thank you for your comment. We agree that the paper, especially the results, can be better written and more focused to allow easier understanding of the important findings.

Comment 3:

The introduction would benefit from a bit more focus on existing Hong Kong data, and how this compares to data from mainland China/Taiwan. The introduction mentions the limitations of existing Hong Kong data, but not what is currently known (while acknowledging those limitations). Given that results are compared with mainland China/Taiwan in the abstract/conclusion, it might be useful to introduce some ideas about why differences in Hong Kong might be expected. Furthermore, given that trends were investigated (and found) in the data, the fact that existing data from China (and elsewhere) was collected at different time points should be acknowledged. Authors could also mention the importance of standardisation in the introduction, given that your results show the need to standardise carefully.

Authors’ reply:

Existing Hong Kong data and the time points of data from other locations were added in introduction section. More explanation of why differences between mainland China/ Taiwan and the importance of standardisation were provided in the discussion section.

Comment 4:

The second paragraph on testing VA and refractive errors initially seemed fairly clear, but actually ultimately I found it quite confusing. I wonder what is the purpose of this section (to make the case for the measure of VA?) and what does this add to the paper. It doesn’t seem to fit the narrative very well. (Perhaps I only say this because I am not an expert on visual health.)

Authors’ reply:

This paragraph was shortened and re-organized in the introduction section.

Comment 5:

One very important piece of information is missing from the methods/results: how many students are covered by the annual health check, and what is this as a percentage of Hong Kong students? The results are based on prevalence, but raw descriptive statistics are missing. (Or perhaps I have missed this, but if so, they should be much more prominent.)

Authors’ reply:

We have added a table and report the total number and participation rate of primary and secondary students and schools.

Comment 6:

Before explaining the use of multiple imputation to handle missing data the researchers should say something about how much data (and which data) is actually missing.

Authors’ reply:

We have added a table to describe the missing data pattern of variables of schoolchildren included in the study.

Comment 7:

I understand that for the purpose of presenting the results, choosing 4 time intervals and 5 age/grade intervals may be helpful, but it would be reassuring to know that these choices do not substantially affect the results.

Authors’ reply:

Since the existing font size in tables were 5 to 8, it is not feasible to use 16 time intervals and 10 age/grade intervals.

Comment 8:

The results for grade seem to be basically the same as for age (which is hardly surprising, I assume they correlate very highly). Unless I am missing a good reason to keep both, I think the paper would be easier to read if the authors just state once that results for age and grade were very similar and then subsequently only write about age, with all further references/results regarding grade only included in supplementary files.

Authors’ reply:

Age and grade were kept in the model since there were confounding and mediation effect according to confounder criteria and Baron and Kenny criteria.

Comment 9:

On a related note, I generally find it useful to report/read correlations between predictors and would encourage the authors to add a table with this information.

Authors’ reply:

We respect your personal preference and suggestions. However, our intention on conducting statistical analysis to test confounding, mediation and interaction, instead of correlation for all variables, stemmed from our aim for better standardization in reporting and compliance to item 7 of the STROBE Checklist for cross-sectional studies (www.strobe-statement.org). We sincerely hope you can understand our intention and accept that there can be differences in preferred reporting style among researchers.

Comment 10:

It took me some time to understand the first paragraph of the results (in particular what the correlation shows, and why it is so strong). I assume what you are trying to say here is that reduced VA prevalence has a very strong negative correlation with SHS attendance/enrolled students because age is a strong predictor of VA? This result might be easier for the reader to understand and interpret if you also explain the crude and age-sex adjusted prevalence rates, and note the differences with the standardized prevalence rate. (Good standardization practice could also be further discussed in the discussion.) Furthermore, it may be useful to provide readers with some explanation of why the proportion of primary to secondary students attending SHS (and enrolled) has changed so much over time.

Authors’ reply:

Thank you for your valuable advice, we have since made the amendments in the Discussion section.

Comment 11

The tables are difficult to read with the present formatting (so I hope authors/editors will ensure that numbers appear on one line wherever possible, e.g., by setting the relevant pages to ‘landscape’ format.) Table 2 moves back and forth between prevalence data (a/c) and odds ratios (b/d) which is difficult to read. If the researchers accept my suggestion to remove grade data, then c/d are not needed, but if they reject the suggestion, then I think this table should be reordered (e.g., acbd).

Authors’ reply:

Thank you for suggestions. We have now divided the original table into two separate tables for easier reading. One table shows the prevalence, while the other shows the odds ratio.

Comment 12:

Prevalence changes over time are one of the main results of the paper, yet I cannot see any statistical testing of whether these changes are significant. Perhaps it is implied by non-overlapping confidence intervals, but the results would be more robust if changes over time were explicitly tested and reported on.

Author’s reply:

Mann-Kendall trend test was performed to show the prevalence changes over time. The results were provided in Table 1 and 2.

Comment 13:

To make the odds ratio, Author should add per year (describe this in text).

Authors’ reply:

Since the existing font size in tables were 5 to 8, it is not feasible to use 16 time intervals and 10 age/grade intervals.

Comment 14:

The statement “implying that the interaction effect between age and sex was statistically significant” seems very odd, should not authors just actually test with the interaction effect was significant? (This interaction is also mentioned in the discussion, but I did not see the test result anywhere.)

Author’s reply:

Yes, agree. We have now added a table to show the test result of interaction effect.

Comment 15:

The results regarding geographical variations (Type of housing, School district, Home district and Student Health Service Centres) are inadequately presented and interpreted. Either significant revision is needed to make these results interpretable to people who do not have much knowledge of Hong Kong geography, or these results should be removed (and potentially made the subject of a separate paper). I suggest removal, because these geographical variations are barely mentioned in the introduction, and in the discussion section the explanations are very speculative and completely unreferenced. (Authors even acknowledge this for one of the predictors mentioned: “Due to the limitation of our existing data [on SHSCs], we could not conduct in-depth investigation and provide further explanation.”) I believe that the paper is still sufficient without these results, there is still important data on time trends, age and gender.

Authors’ reply:

Due to the limitation of our retrospective study, the raw data of the Student Health Service did not allow us to perform in-depth analysis of sociological effects on the prevalence of reduced VA at person-level. Type of housing would partially but not totally represent the social class in Hong Kong. In addition, we planned to conduct a multilevel analysis in another study to investigate how the compositional and contextual factors affect the prevalence of reduced VA and explain the causes of geographical variations by combining the data at district-level from other official departments such as median monthly household income, household size, population density and greenery scale.  

Comment 16:

The discussion needs a lot of work, both on structure and content. Authors should start by clearly laying out their general findings, and only then explaining what these findings add to the existing literature, and how they compare with mainland China/Taiwan etc.

Authors’ reply:

Thank you for your suggestion, we have now amended the Discussion section.

Comment 17:

 Authors should include a section on strengths and limitations.

Authors’ reply:

Agree, a section on strengths and limitations have now been added.

Comment 18:

New terms such as myopia are added to the discussion, which also doesn’t help with readability.

Authors’ reply:
Agree, a detailed explanation has now been added to explained the relationship between myopia, refractive errors and reduced VA.

Comment 19:

There is insufficient explanation and interpretation of the interesting findings regarding changes in the effects of gender and age on VA (and the interaction). Possible explanations should be referenced and compared with existing literature.

Authors’ reply:

Thank you for your suggestion, We have added a table to show the test result, as well as discussed further on possible reason for higher incidence of reduced VA among older school girls.

Comment 20:

The use of ‘educational attainment’ to refer to year of schooling needs amending or explaining – are these results not just age effects (see also comment 8)? Or are they really about educational attainment?

Author’s reply:

We have amended the term of “educational attainment” to “grade of students”.

Comment 21:

Authors should be very wary of introducing new results (on cross-boundary students) in the discussion. I can see why they may be relevant to the overall trends found in the paper, but I think this idea should be included in the introduction (and thus justifying the including of relevant results in the results section and subsequent discussion).

Authors’ reply:

We have mentioned why the cross-boundary students is relevant in the Introduction section.

Comment 22

The conclusion should not introduce new ideas (such as ‘the severity of refractive errors’).

Authors’ reply:

We have relocated it to the Discussion section.

Reviewer 2 Report

The manuscript presents a significant amount of data that requires further discussion, particularly in the introduction.

The intended geographical approach is lost among the information. Accuracies are required on what are the real geographical possibilities to explain a public health issue such as the one developed in the manuscript.

The variables of living environment and social classes, in addition to Geographic distribution and variation have a greater weight in the preparation of the results. The central question of the work towards this direction could be redirected. Observe the relevance of Type of housing for VA. There are locational elements that can contribute to a more complex reading of the problem.

In general, it is a good job with abundant data. The latter being positive, you should concentrate more on solving aspects that the numbers alone do not explain.

Author Response

Reviewer 2

Comment 1:

The manuscript presents a significant amount of data that requires further discussion, particularly in the introduction.

Author’s reply:

Thank you for your comment. For your information, the data in the introduction were used to illustrate the associations between refractive errors and reduced VA, and between myopia and reduced VA. The purpose was to show their relationship.

Comment 2:

The intended geographical approach is lost among the information. Accuracies are required on what are the real geographical possibilities to explain a public health issue such as the one developed in the manuscript.

Authors’ reply:

We agree. We have now added a table to show Moran’s Index and Mantel test to investigate the spatial autocorrelation and described this in text. In addition, we have performed chi-square independence test and Kruskal–Wallis test to show the difference of prevalence and odds ratio for each school or home district.

Comment 3:

The variables of living environment and social classes, in addition to Geographic distribution and variation have a greater weight in the preparation of the results. The central question of the work towards this direction could be redirected. Observe the relevance of Type of housing for VA. There are locational elements that can contribute to a more complex reading of the problem.

Author’s reply:

Due to the limitation of our retrospective study, the raw data of the Student Health Service did not allow us to perform in-depth analysis of sociological effects on the prevalence of reduced VA at person-level. Type of housing would partially but not totally represent the social class in Hong Kong. In addition, we planned to conduct a multilevel analysis in another study to investigate how the compositional and contextual factors affect the prevalence of reduced VA and explain the causes of geographical variations by combining the data at district-level from other official departments such as median monthly household income, household size, population density and greenery scale.  

Comment 4:

In general, it is a good job with abundant data. The latter being positive, you should concentrate more on solving aspects that the numbers alone do not explain.

Authors’ reply:

Thank you for your advice. We have now made further amendments for improving this article.

Reviewer 3 Report

The methods of VA measurement are important and are not mentioned. The authors should at least reference the methods to measure VA. In addition, the relevance of myopia is of high significance in this population and therefore the subjects should be stratified on their refractive error subgroup . A breakdown of the percentage of subjects analyzed in this study who are representative of emmetropia, hypermetropia, myopia should be presented.  The  results section is very tedious. Authors should consider scatter gram to represent data in the tables which correlate age /sex as well as refractive error /age. These would be more valuable to the reader than enormous amount of data in the table. 

Finally, this work is an important awareness effort and more discussion should be analyzed as far as the sociological factors of influence on why the district distribution of data demonstrated the results within this study group. This is a very compelling conclusion that the authors could have expounded on further.Please also consider references further to the sociological impact factors which effect the district performance measures. this is significant when you isolate regions of variance.

Author Response

Reviewer 3

Comment 1:

The methods of VA measurement are important and are not mentioned. The authors should at least reference the methods to measure VA.

Authors’ reply:

Thank you for your comments. For your information, we have already described the method of measuring VA and the definition of reduced VA in the paragraph of Definitions of MATERIALS and METHODS (on Page 2).

Comment 2:

In addition, the relevance of myopia is of high significance in this population and therefore the subjects should be stratified on their refractive error subgroup. A breakdown of the percentage of subjects analyzed in this study who are representative of emmetropia, hypermetropia, myopia should be presented. 

Authors’ reply:

Due to the limitation of our study, we checked the VA but did not measure the refractive errors, i.e. we did not have the figures of emmetropia, hypermetropia and myopia. Since it is costly to adopt large-scale cycloplegic refraction covering most schoolchildren, our result was useful to predict the overall prevalence of refractive errors and myopia in Hong Kong and compare our estimates with other randomized sampling studies.  

Comment 3:

The results section is very tedious. Authors should consider scatter gram to represent data in the tables which correlate age /sex as well as refractive error /age. These would be more valuable to the reader than enormous amount of data in the table. 

Authors’ reply:

Thank you for your comments. We have now made further changes. The significant differences and correlations are now bolded and more tables have been added to explain the correlation of age and sex, as well as refractive error and age.

Comment 4:

Finally, this work is an important awareness effort and more discussion should be analyzed as far as the sociological factors of influence on why the district distribution of data demonstrated the results within this study group. This is a very compelling conclusion that the authors could have expounded on further. Please also consider references further to the sociological impact factors which effect the district performance measures. this is significant when you isolate regions of variance.

Author’s reply:

Due to the limitation of our retrospective study, the raw data of the Student Health Service did not allow us to perform in-depth analysis of sociological effects on the prevalence of reduced VA at person-level. Type of housing would partially but not totally represent the social class in Hong Kong. In addition, we planned to conduct a multilevel analysis in another study to investigate how the compositional and contextual factors affect the prevalence of reduced VA and explain the causes of geographical variations by combining the data at district-level from other official departments such as median monthly household income, household size, population density and greenery scale.   

Reviewer 4 Report

The conclusion should be written more transparent&

Every significant result and tendency should get an explanation and lead to a certain point in conclusion

The significant differences and correlations could be underlined for better detection by reader

Author Response

Reviewer 4

Comment 1:

The conclusion should be written more transparent&

Authors’ reply:

Thank you for the suggestion, we have now re-written the conclusion.

Comment 2:

Every significant result and tendency should get an explanation and lead to a certain point in conclusion

Authors’ reply:

Agree, explanation for every significant results and tendencies have now been added.

Comment 3:

The significant differences and correlations could be underlined for better detection by reader

Authors’ reply:

Agree, we have now amended the manuscript so that all significant differences and correlations are now bolded.

Round 2

Reviewer 1 Report

General Comments

I appreciate that substantial work has been done to improve the manuscript and can see improvement across several areas. However, I still have general concern that the interesting findings of the study are not clearly communicated, likely meaning that readership and use of the paper would be lower than it could be. For example, I appreciate the author’s transparency in sharing results, but the presence of 13 tables in the main text is an indication that not enough work has been done by authors to determine the key findings, summarise them, and communicate them clearly. See some suggestions for what could be moved to supplementary information below, though I also hope the authors may decide on further revisions of this type.

Abstract

In several places the language in the abstract is unclear, I suggest: “Girls were less susceptible than boys at age 6-7 (and in grade primary 1-2), but more susceptible at older ages. The prevalence in junior grades increased while the risk effect of grade reduced over the past 17 years.” I don’t understand what is meant by “spatial autocorrelation… existed” – do you mean positive or negative autocorrelation (or both)? This is a new term to me, but as I understand it, saying it exists is not very meaningful without a direction. “Multilevel research should be conducted to review the influence of compositional and contextual factors on reduced VA.”

Introduction

I would prefer to see the information in the first two paragraphs more succinctly summarised. Detailed results are more distracting than helpful, authors could merely state something like “Studies this century in Vietnam, India and Malaysia have found that refractive error accounted for between 80 and 96% of reduced VA.” In a similar way the results from China and the myopia results (paragraph 2) could also be reduced to much shorter sentences. I like the point that “Third, they estimated the prevalence at a single time point but did not describe the dynamic changes over a series of time points…” but do not find the subsequent reference to cross-boundary students helpful in its current state. I think it would be better to say something like “… despite social and demographic trends (such as the numbers of cross-boundary students) that may have led to changes in prevalence over time”.

Methods and Results

I know that I asked for the information now presented in Table 1 and 2 (demographics and missing data), but I recommend that authors include these tables as supplementary information. Furthermore, simply adding the Total N (millions?) clearly in the results would reassure readers that these results are based on a very large and impressive dataset! I am not familiar with the Mann-Kendall trend test, Moran’s Index and the Mantel test, so I hope other reviewers can determine if their use is appropriate. Given that other readers may also be new to these tests, the results (p13) could perhaps make it a bit clearer what significant test results mean. It is good to see the interaction results, but I am very concerned about their portrayal – for example the way you interpret the ORs in Table 6 for age and sex. (This should have been noticed already, because you now state that “girls was associated with lower risk of reduced VA”, in contrast to your earlier findings!)

In the interaction model, the OR of .795 is the effect of girls when age = 0 (which is pretty meaningless for your study – this webpage is one of many resources on the topic). If you centre (or otherwise rescale – perhaps around age 6, the youngest age) your variables, then the coefficients for the main effects may be more meaningful. I realise that interpreting interactions when conducting logistic regression is challenging, so I suggest considering further guidance on this. Perhaps this webpage is a good start.

I generally agree with your interpretation that the effect of girls on reduced VA becomes more negative with age but I am not sure about calling it a ‘beneficial effect’.

Some of the language in the new section on age and sex is also unclear, I suggest: “being a girl [/female] was associated...” “modify the effect of” rather than “effect modifier”. I still do not really understand the school type and grade results, nor your point in your response letter that ‘Age and grade were kept in the model since there were confounding and mediation effect according to confounder criteria and Baron and Kenny criteria’. The results (or perhaps your presentation of them - the new section is also very sloppily written, e.g., grater/medication) does not yet convince me that you understand the effect of grade (or school-type) over and above the effect of age (which seems to me to be the ‘main’ result).

(Perhaps there is something that I don’t understand about the Hong Kong school system, because in most school systems I know of grade and age will be very highly correlated (.9), so including both analyses just doesn’t make sense to me. Every time you mention a grade/school-level result/effect, I think, “This is probably just an age effect”. This is one reason I suggested adding correlations between predictors. I do not really see why this is incompatible with item 7 of the STROBE Checklist.)

My recommendation remains that you do not include grade in your main analyses. What does it add at present, other than substantial confusion? This study is not the place to look into causal/mediational relationships between age, grade, school-type and your outcome. (Not least because you cannot establish causation with cross-sectional data.)

I can see a bit more rationale for including school-type (your finding that the ORs of primary school compared to secondary change are interesting) but I still wonder what this means over and above an age-effect, and your discussion doesn’t make this any clearer.

Discussion

The discussion is much improved, but still strikes slightly the wrong tone from the start. First summarise the main findings – the results outlined in the abstract. The paragraph on primary/secondary student prevalence should be used to explain these results (and their difference from previous results), rather than open the discussion. (I find Larry Steinberg’s rules for discussion writing helpful, though perhaps the authors think things are different in their discipline.) The following sentence is not clear: “Due to the limitation of retrospective data and the scope of our study, we would include this factor in multivariate logistic analysis to control its potential effect but would not conduct in-depth investigation and provide further explanation of the variation.” There are further English mistakes in new sections (e.g., “this finding was consistence with previous studies”) so please review.

Author Response

A serial cross-sectional analysis of the prevalence, risk factors and geographic variations of reduced visual acuity in primary and secondary students from 2000 to 2017 in Hong Kong.

Version 2

Reviewer Comments to Author

General Comments

I appreciate that substantial work has been done to improve the manuscript and can see improvement across several areas. However, I still have general concern that the interesting findings of the study are not clearly communicated, likely meaning that readership and use of the paper would be lower than it could be. For example, I appreciate the author’s transparency in sharing results, but the presence of 13 tables in the main text is an indication that not enough work has been done by authors to determine the key findings, summarise them, and communicate them clearly. See some suggestions for what could be moved to supplementary information below, though I also hope the authors may decide on further revisions of this type.

Authors’ reply:

We have moved 8 of the 13 tables to the supplementary information section, so only 5 tables and 2 figures now remain in the main text.

Abstract

In several places the language in the abstract is unclear, I suggest: • “Girls were less susceptible than boys at age 6-7 (and in grade primary 1-2), but more susceptible at older ages. The prevalence in junior grades increased while the risk effect of grade reduced over the past 17 years.” I don’t understand what is meant by “spatial autocorrelation… existed” – do you mean positive or negative autocorrelation (or both)? This is a new term to me, but as I understand it, saying it exists is not very meaningful without a direction. “Multilevel research should be conducted to review the influence of compositional and contextual factors on reduced VA.” Authors’ reply: We agreed with the reviewer’s suggestion and hae made the appropriate amendments.

Introduction

I would prefer to see the information in the first two paragraphs more succinctly summarised. Detailed results are more distracting than helpful, authors could merely state something like “Studies this century in Vietnam, India and Malaysia have found that refractive error accounted for between 80 and 96% of reduced VA.” In a similar way the results from China and the myopia results (paragraph 2) could also be reduced to much shorter sentences. Authors’ reply: We agreed with the reviewer’s suggestion and have made the appropriate amendments.

I like the point that “Third, they estimated the prevalence at a single time point but did not describe the dynamic changes over a series of time points…” but do not find the subsequent reference to cross-boundary students helpful in its current state. I think it would be better to say something like “… despite social and demographic trends (such as the numbers of cross-boundary students) that may have led to changes in prevalence over time”. Authors’ reply: We agreed with the reviewer’s suggestion and have made the appropriate amendments.

Methods and Results

I know that I asked for the information now presented in Table 1 and 2 (demographics and missing data), but I recommend that authors include these tables as supplementary information. Furthermore, simply adding the Total N (millions?) clearly in the results would reassure readers that these results are based on a very large and impressive dataset!

Authors’ reply:

Thank you for your suggestion, we have now moved Table 1 and 2 to the supplementary information. The total frequency of schoolchildren meeting the inclusion criteria have also been added.

I am not familiar with the Mann-Kendall trend test, Moran’s Index and the Mantel test, so I hope other reviewers can determine if their use is appropriate. Given that other readers may also be new to these tests, the results (p13) could perhaps make it a bit clearer what significant test results mean.

Authors’ reply:

The interpretations of the 3 test results have now been amended.

It is good to see the interaction results, but I am very concerned about their portrayal – for example the way you interpret the ORs in Table 6 for age and sex. (This should have been noticed already, because you now state that “girls was associated with lower risk of reduced VA”, in contrast to your earlier findings!)

In the interaction model, the OR of .795 is the effect of girls when age = 0 (which is pretty meaningless for your study – this webpage is one of many resources on the topic). If you centre (or otherwise rescale – perhaps around age 6, the youngest age) your variables, then the coefficients for the main effects may be more meaningful. I realise that interpreting interactions when conducting logistic regression is challenging, so I suggest considering further guidance on this. Perhaps this webpage is a good start.

I generally agree with your interpretation that the effect of girls on reduced VA becomes more negative with age but I am not sure about calling it a ‘beneficial effect’.

  Authors’ reply: Thank you for your valuable suggestions, we have now re-written the interpretation of interaction results.   Some of the language in the new section on age and sex is also unclear, I suggest: • “being a girl [/female] was associated...” “modify the effect of” rather than “effect modifier” Authors’ reply: We agreed with the reviewer’s suggestion and have made the appropriate amendments.   I still do not really understand the school type and grade results, nor your point in your response letter that ‘Age and grade were kept in the model since there were confounding and mediation effect according to confounder criteria and Baron and Kenny criteria’. The results (or perhaps your presentation of them - the new section is also very sloppily written, e.g., grater/medication) does not yet convince me that you understand the effect of grade (or school-type) over and above the effect of age (which seems to me to be the ‘main’ result). (Perhaps there is something that I don’t understand about the Hong Kong school system, because in most school systems I know of grade and age will be very highly correlated (.9), so including both analyses just doesn’t make sense to me. Every time you mention a grade/school-level result/effect, I think, “This is probably just an age effect”. This is one reason I suggested adding correlations between predictors. I do not really see why this is incompatible with item 7 of the STROBE Checklist.) My recommendation remains that you do not include grade in your main analyses. What does it add at present, other than substantial confusion? This study is not the place to look into causal/mediational relationships between age, grade, school-type and your outcome. (Not least because you cannot establish causation with cross-sectional data.)

I can see a bit more rationale for including school-type (your finding that the ORs of primary school compared to secondary change are interesting) but I still wonder what this means over and above an age-effect, and your discussion doesn’t make this any clearer.

Authors’ reply

Large relative changes in ORs of grade (and school type) on reduced VA indicated the presence of the confounding effect of age on influencing the observed association between grade (and school type) and reduced VA in univariate and multivariate analysis. The adjusted measure gave a better correlation and estimate than using the crude measure, since it disregarded the confounding effect of age. Without this adjustment, a negative confounding effect would result in an underestimation of the risk effect of grade during 2000/01-2003/04, and inversely, a positive confounding effect would overestimate the risk effect of grade from 2004/05 to 2016/17 and the risk effect of school type from 2000/01 to 2016/17. The rising relative change in ORs showed that the extent of overestimating the association between grade (and school type) and reduced VA had enlarged.

Since the confounding effect reflected the natural relationships between lifestyle, habits and other characteristics, we expect age might possibly be associated with other potential risk factors for reduced VA which are not covered in our study. The risk effect of age is not completely equivalent to grade. 

 (Reference: http://sphweb.bumc.bu.edu/otlt/MPH-Modules/BS/BS704-EP713_Confounding-EM/BS704-EP713_Confounding-EM3.html)

In addition, in multivariate analysis, there was significantly estimate of grade of students in multivariate analysis so that with the same background information, senior students were assumed to have higher risk for reduced VA than junior students at the same age. Similarly, older students were assumed to have higher risk for reduced VA than younger students at the same grade. If we include either age or grade in the multivariate analysis, the model could not reflect this observation.     

Regarding the school system in Hong Kong, primary school is equivalent to P1-P6 while secondary School is equivalent to S1-S4. In other words, P1-P6 was subgroups of primary school whereas S1-S4 is the subgroups of secondary school. Thus, we mostly used grade of students for analysis instead of school type.

Discussion

The discussion is much improved, but still strikes slightly the wrong tone from the start. First summarise the main findings – the results outlined in the abstract. The paragraph on primary/secondary student prevalence should be used to explain these results (and their difference from previous results), rather than open the discussion. (I find Larry Steinberg’s rules for discussion writing helpful, though perhaps the authors think things are different in their discipline.)

The following sentence is not clear: “Due to the limitation of retrospective data and the scope of our study, we would include this factor in multivariate logistic analysis to control its potential effect but would not conduct in-depth investigation and provide further explanation of the variation.”

Authors’ reply:

We have amended the Discussion accordingly and removed the sentence mentioned by point 17., to avoid any confusion.

There are further English mistakes in new sections (e.g., “this finding was consistence with previous studies”) so please review.

Authors’ reply

Thank you for your comment, we have now made further attempts to remove any remaining grammatical errors as much as possible.
